# Group II intron inhibits conjugative relaxase expression in bacteria by mRNA targeting

**Guosheng Qu[1,2]\*, Carol Lyn Piazza[1,2], Dorie Smith[1,2], Marlene Belfort[1,2,3]\***

[1]Department of Biological Sciences, University at Albany, Albany, United States; [2]RNA Institute, University at Albany, Albany, United States; [3]Department of Biomedical Sciences, School of Public Health, University at Albany, Albany, United States

**Abstract** Group II introns are mobile ribozymes that are rare in bacterial genomes, often cohabiting with various mobile elements, and seldom interrupting housekeeping genes. What accounts for this distribution has not been well understood. Here, we demonstrate that Ll.LtrB, the group II intron residing in a relaxase gene on a conjugative plasmid from *Lactococcus lactis*, inhibits its host gene expression and restrains the naturally cohabiting mobile element from conjugative horizontal transfer. We show that reduction in gene expression is mainly at the mRNA level, and results from the interaction between exon-binding sequences (EBSs) in the intron and intron-binding sequences (IBSs) in the mRNA. The spliced intron targets the relaxase mRNA and reopens ligated exons, causing major mRNA loss. Taken together, this study provides an explanation for the distribution and paucity of group II introns in bacteria, and suggests a potential force for those introns to evolve into spliceosomal introns.

DOI: https://doi.org/10.7554/eLife.34268.001

**\*For correspondence:**
guosheng.qu@yahoo.com (GQ);
mbelfort@albany.edu (MB)

**Competing interests:** The authors declare that no competing interests exist.

## Introduction

Introns interrupt genes in all domains of life. Group II introns, which are found in bacterial and organellar genomes, are ribozymes that self-splice from pre-mRNA transcripts independent of protein catalysis (*Lambowitz and Belfort, 2015*; *Lambowitz and Zimmerly, 2011*; *Zimmerly and Semper, 2015*). Splicing of group II introns occurs by the same pathway as that of spliceosomal introns, of which group II introns are the presumed ancestors (*Lambowitz and Belfort, 2015*; *Novikova and Belfort, 2017*; *Zimmerly and Semper, 2015*). Group II introns are also mobile retroelements that transpose into DNA via an RNA intermediate (*Lambowitz and Zimmerly, 2011*). Both splicing and retromobility of group II introns in vivo require an intron-encoded protein (IEP) that is in complex with the intron (*Matsuura et al., 2001*; *Matsuura et al., 1997*; *Qu et al., 2016*; *Wank et al., 1999*; *Zimmerly et al., 1995*).

Group II introns have a highly conserved RNA structure consisting of six domains (*Qu et al., 2016*; *Robart et al., 2014*; *Toor et al., 2008*). The largest domain, DI, contains the exon-binding sequences (EBS), which interact through base pairing with the intron-binding sequences (IBS) in the RNA exons or DNA homing target, to define the sites for splicing and retromobility of the intron, respectively (*Lambowitz and Zimmerly, 2011*). DV is structurally the most conserved domain, which contains the 'catalytic triad', nucleotide residues critical for catalysis (*Qu et al., 2016*; *Robart et al., 2014*; *Toor et al., 2008*). DVI has a bulged adenosine, which is the nucleophile that initiates the splicing reaction, yielding the branch-point of an intron lariat resulting from splicing. DIV often accommodates an open reading frame (ORF) encoding the IEP, which is a reverse transcriptase (RT) with maturase activity that facilitates intron splicing (*Matsuura et al., 2001*; *Matsuura et al., 1997*;

**eLife digest** Proteins, the molecular workhorses of the cell, are encoded by genes within the DNA. A given gene is often formed of several portions of coding DNA, the exons, which are separated by sections of non-coding DNA called introns. When it is expressed, the entire gene, including the introns, is 'copied' into a molecule of pre-messenger RNA (pre-mRNA). Enzymes then remove the introns and stitch the exons together to produce a mature messenger RNA (mRNA) that can serve as a template to create a protein. However, a particular type of intron, known as a group II intron, can remove or 'splice' itself out of a pre-mRNA without the help of additional enzymes. Once this is done, these introns can also insert themselves into DNA.

Scientists think that group II introns originated in bacteria, yet only about a quarter of bacterial genomes sequenced so far contain this particular kind of intron. When these introns are present, their numbers are low, and most of the time they are restricted to genes that are not required for survival. It is not well understood why group II introns are so few and far between, and why they are often not associated with essential genes in bacteria.

Qu et al. looked into a gene that contains a group II intron in the bacterium *Lactococcus lactis*, and discovered that the expression of this gene was dramatically low. This decrease was due to interactions between the group II intron and the exons within the mRNA. Once group II introns were spliced out, they could target their mRNAs and reopen the junction where the exons had been stitched together. These mRNAs were therefore lost, and the cell made less of the protein that the gene encoded.

The findings by Qu et al. can help to explain why group II introns are so scarce, how they are excluded from essential genes, and also how cells could use these sequences. In particular, group II introns could have evolved to form the spliceosome, a complex structure that processes introns in higher organisms. Finally, group II introns could be used in the laboratory to artificially silence genes of interest.

DOI: https://doi.org/10.7554/eLife.34268.002

Wank et al., 1999; Zimmerly et al., 1995), and can promote retromobility of the intron in combination with the DNA endonuclease activity of the IEP (Matsuura et al., 1997; Qu et al., 2016; Zimmerly et al., 1995).

Distribution of group II introns is different in bacteria and eukaryotes. Only about one quarter of sequenced bacterial genomes contain group II introns and they are usually present in small numbers within one genome. Moreover, they are mostly located in intergenic regions or non-conserved genes that are rarely essential. In addition, bacterial group II introns frequently reside in various types of mobile DNAs, which include pathogenicity islands and virulence plasmids (Candales et al., 2012; Dai et al., 2003; Toro and Martínez-Abarca, 2013). This distribution suggests that bacterial group II introns might be deleterious to host gene expression (Dai and Zimmerly, 2002a; Zimmerly and Semper, 2015). In contrast, organellar introns are distributed more densely in mitochondrial and chloroplast genomes, and are almost exclusively in essential housekeeping genes. In addition, group II introns are absent in eukaryotic nuclear genomes although they are ancestrally closely related to spliceosomal introns (Lambowitz and Belfort, 2015; Novikova and Belfort, 2017; Zimmerly and Semper, 2015). Factors, including nucleus-cytosol compartmentalization, intracellular magnesium concentrations, and interactions between the intron and spliced mRNA, have been conjectured to be the evolutionary drivers for the evolution of group II introns into spliceosomal introns in eukaryotes (Martin and Koonin, 2006; Qu et al., 2014; Truong et al., 2015).

Here we investigated the impact of group II introns on their host gene expression in bacteria, using Ll.LtrB, the group II intron that resides on a conjugative plasmid pRS01 and interrupts a relaxase host gene *ltrB*, in the bacterium *Lactococcus lactis* (Mills et al., 1996). We demonstrate that this intron inhibits relaxase host gene expression and mobilization of the cohabiting conjugative element. This group II intron-promoted inhibition results from RNA-RNA interactions between the spliced intron and mRNA, reducing mRNA levels. These discoveries can explain the distribution of group II introns in bacteria and provide yet another example of strict control and silencing of conjugation

(*Singh and Meijer, 2014*; *Zatyka and Thomas, 1998*). These data also suggest a driving force for the evolutionary transition of the group II introns to spliceosomal introns.

## Results

### A group II intron inhibits gene expression at the mRNA level

The group IIA Ll.LtrB intron interrupts the *ltrB* relaxase gene, which resides on the conjugative plasmid pRS01 and is required for pRS01 conjugative transmission (*Belhocine et al., 2004*; *Belhocine et al., 2005*; *Mills et al., 1996*). Both intron-containing (Int[+]) and intron-less (Int[-]) *ltrB* genes, under control of a nisin-inducible promoter on a pCY20 plasmid, were expressed in the native host *L. lactis* strain IL1403 (*Figure 1A*). Northern blotting analysis of RNA (*Figure 1B*) indicated that the relaxase mRNA abundance was much lower in the presence of the intron than in its absence (21% vs. 100%). Consistent with the mRNA difference, the LtrB relaxase determined by Western blotting also accumulated to lower levels in the Int[+] cell (*Figure 1C*, 36% vs. 100%). Notably, a previous study showed an even larger (20-fold) difference in protein levels in Int[-] versus Int[+] cells (*Chen et al., 2005*). The dramatic decrease in the *ltrB* relaxase mRNA level in the presence of the intron was confirmed by performing quantitative reverse transcription PCR (qRT-PCR). The difference in mRNA levels between the Int[+] and the Int[-] cells was 23-fold (*Figure 1D*, *Figure 1—source data 1*, *Figure 1—figure supplement 1*, 4.3% vs. 100%). The greater difference by qRT-PCR and Northern blotting, may be accounted for by the different reference RNAs used (copA mRNA for qRT-PCR vs. 16S rRNA for Northern blot) or by the inherent difference between the two techniques in determining RNA levels.

By performing both reverse transcription and qRT-PCR analyses we determined that the decrease in mRNA was not simply due to group II intron-promoted reduction in transcription rate (*Figure 1—figure supplement 1*). We analyzed nascent *ltrB* primary transcripts in Int[-]/Int[+] cells, utilizing a DNA primer that bound 50 nucleotides downstream from the 5′-end of the transcript (*Figure 1—figure supplement 1A–B*). Because of the proximity of the primer to the 5′-end, we presumed that the yield of the cDNA or RT-PCR products generated was a reflection of transcription initiation, and indeed there was only a 20% difference in the amount of cDNAs between Int[-] and Int[+] (*Figure 1—figure supplements 1A*, 100% vs. 79%; **1B**, 100% vs. 74%). In addition, comparison of the relative mRNA expression level in the Int[-] cell with that of total intron RNAs (pre-mRNA + Intron) in the Int[+] cell, also indicated small transcription differences between Int[-]/Int[+] cells (*Figure 1—figure supplement 1C*, primers, **1D**, relative levels, 20.04 ± 2.00 vs. 25.65 ± 6.47).

In regard to splicing, we used Northern blotting, primer extension and qRT-PCR to determine the relative abundance of the spliced intron. The data showed that the sum of intron-containing RNAs (pre-mRNA + Intron) was relatively abundant, in contrast to the spliced mRNA (*Figure 1B*, *Figure 1—figure supplement 1D*). Notably, a previous analysis of Ll.LtrB group II intron expression in the natural pRS01 context also showed that the *ltrB* mRNA level was much lower than that of the total intron RNA (*Chen et al., 2005*). When the intron splicing efficiency was defined as the ratio of spliced intron against the sum of the intron-containing RNAs (pre-mRNA + Intron), the numbers ranged from 63% to 91% (*Figure 1B*, 91%; *Figure 1—figure supplement 1D*, 73.8%; *Figure 1—figure supplement 2B*, 63%). Thus, it does not appear that a splicing deficiency alone resulted in the mRNA reduction.

Furthermore, we compared the decay rate of mRNAs expressed in Int[-]/Int[+] cells. After a 30 min induction of *ltrB* gene expression, the cells were treated with rifampicin to stop transcription, and mRNA level was monitored over time and analyzed by Northern blotting (*Figure 1—figure supplement 3A–B*). Surprisingly, the decay rate of spliced mRNA for the Int[+] cells was ¼ to ½ of the rate for the mRNA of the Int[-] cells. Because the rates apply to a small residual fraction of the mRNA, these results must be interpreted with caution. Nevertheless, they suggest that the reduction in mRNA is not due to increased degradation.

We also investigated if gene expression was affected at the translation level. Comparison of mRNA polysome profiling showed that there was only a marginal difference in mRNAs enriched in the polysomal fractions between Int[-]/Int[+] cells (*Figure 1—figure supplement 4*, 100% vs. 85%), thereby rendering unlikely the possibility that the spliced mRNA experienced translational repression.

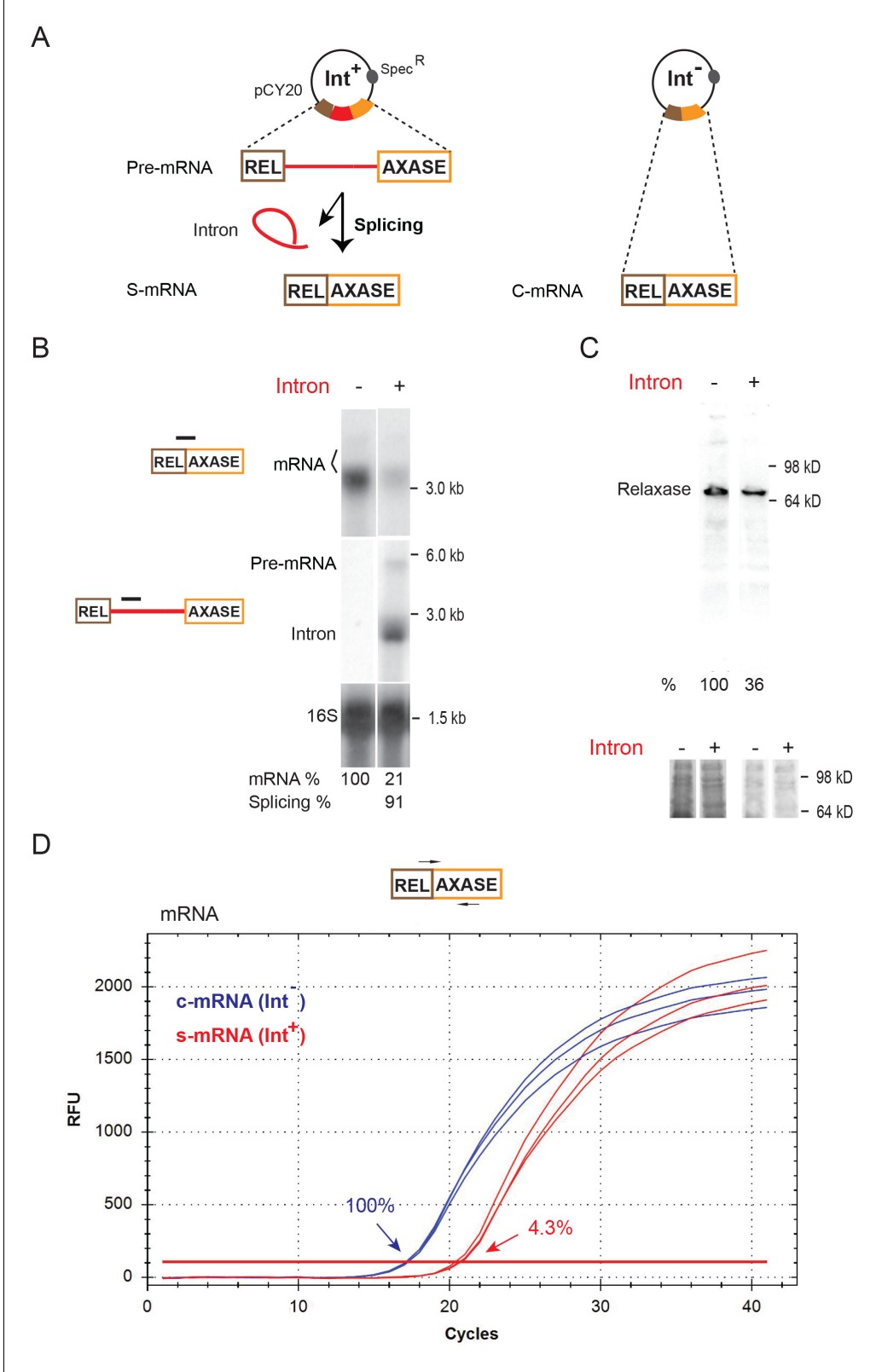

**Figure 1.** Inhibition of host gene expression by Ll.LtrB group II intron. (**A**) Diagram of the *ltrB* relaxase mRNA produced from full-length intron-containing pre-mRNA and intron-less constructs on the pCY20 plasmids (*spec$^R$*). The mRNA generated from splicing of the intron (red) (S-mRNA, left) is compared to mRNA expressed from the intron-less construct (C-mRNA, right). (**B**) RNA analysis by Northern blotting using the mRNA splice-junction and intron-specific probes. Representative data of three biological replicates is shown. Quantitation of mRNA (two bands, bracketed) and splicing,

*Figure 1 continued on next page*

*Figure 1 continued*

normalized to 16S rRNA from the same blot after stripping and reprobing, is denoted at the bottom of the blot. The faint upper mRNA band may result from an alternate transcription start site. Splicing efficiency is defined as the percentage of the spliced intron relative to the sum of pre-mRNA and spliced intron. (C) LtrB relaxase protein levels determined by Western blotting. Representative data of three biological replicates is shown. Quantitation of protein level, normalized to respective total protein levels, is denoted at the bottom of the blot. Portions of the coomassie stained gels before (left) and after transfer (right) are shown below. (D) Representative qRT-PCR profile of expressed mRNAs in the Int⁻ (blue) and Int⁺ (red) cells. PCR target and primer pair are shown (top) and average relative mRNA levels in Int⁻/Int⁺(100% vs 4.3%) derived from mean $C_t$ values of three biological replicates are indicated.

DOI: https://doi.org/10.7554/eLife.34268.003

The following source data and figure supplements are available for figure 1:

**Source data 1.** RT-qPCR data for mRNA.

DOI: https://doi.org/10.7554/eLife.34268.008

**Figure supplement 1.** RNA analysis by qRT-PCR and reverse transcription.

DOI: https://doi.org/10.7554/eLife.34268.004

**Figure supplement 2.** Determination of splicing efficiency by reverse transcription analyses.

DOI: https://doi.org/10.7554/eLife.34268.005

**Figure supplement 3.** Intron may slow down interacting mRNA turnover.

DOI: https://doi.org/10.7554/eLife.34268.006

**Figure supplement 4.** Polysomal profiling.

DOI: https://doi.org/10.7554/eLife.34268.007

Additionally, although *ltrB* RNA levels can be condition-dependent (*Chen et al., 2005*), the differential expression levels between Int⁺ and Int⁻ genes were not influenced by stressors and environmental changes, such as temperature, pH, salt and redox (data not shown). Taken together, we concluded that the group II intron-promoted inhibition in gene expression is likely to be mainly at the mRNA level.

## Suppressing gene expression in bacteria is a common property of group II introns

We also investigated if this substantial group II-intron mediated decrease in mRNA was independent of plasmids and promoters from which the *ltrB* relaxase gene was expressed (*Figure 2A*). When the Int⁻/Int⁺ *ltrB* genes were expressed from different plasmids, including pDL278 and pAMJ328, where gene expression was driven by either constitutive or pH-inducible promoters, respectively, we observed similar levels of mRNA reduction in the presence of the intron (pDL: 100% vs. 28%; pAMJ: 100% vs. 45%) (*Figure 2A*). This demonstrates that group II intron-promoted inhibition was independent of the plasmid and promoter from which the *ltrB* relaxase gene was expressed.

In addition, we extended this comparative study to other types of bacterial group II introns, including a typical IIB intron, EcI5 (*Dai and Zimmerly, 2002b*) and a IIC intron, BhI1 (*Candales et al., 2012*). The Int⁻/Int⁺ constructs of these introns were fused to an enhanced GFP reporter at the N-terminus. When expressed in *Escherichia coli*, levels of the mRNA dropped substantially for both EcI5 and BhI1 in the presence of the intron (*Figure 2B,C*). Also the GFP reporter was greatly reduced for EcI5 with the intron present (*Figure 2B* bottom). Therefore, it appears that inhibiting gene expression in bacteria is a common property of group II introns.

## Ll.LtrB intron reduces conjugation of pRS01

Because *ltrB* relaxase is required for initiation of plasmid conjugation (*Belhocine et al., 2004*; *Belhocine et al., 2005*; *Mills et al., 1996*), and because *ltrB* gene expression is reduced in the presence of the intron, we wished to determine if the intron may inhibit conjugal transfer of pRS01. To test this hypothesis, we measured conjugation frequency where the donor strain, IL1403, co-hosted erythromycin resistant (erm^R) conjugative plasmid pRS01 *ltrB*⁻ (ΔLtrB::tet) and the pCY20 Int⁻ or Int⁺ plasmid. After mating with fusidic acid-resistant (FA^R) ILI403 the frequency of the Erm^RFA^R exconjugants was measured (*Figure 3*). Consistent with the LtrB relaxase protein levels (*Figure 1C*), the conjugation frequency of the Int⁺ donor was 3- to 4-fold lower that of the Int⁻, at $2.0 \pm 0.1 \times 10^{-4}$ versus $7.3 \pm 10^{-4}$ exconjugants per donor (n = 2, p<0.05) (*Figure 3* and data not shown). This result ties

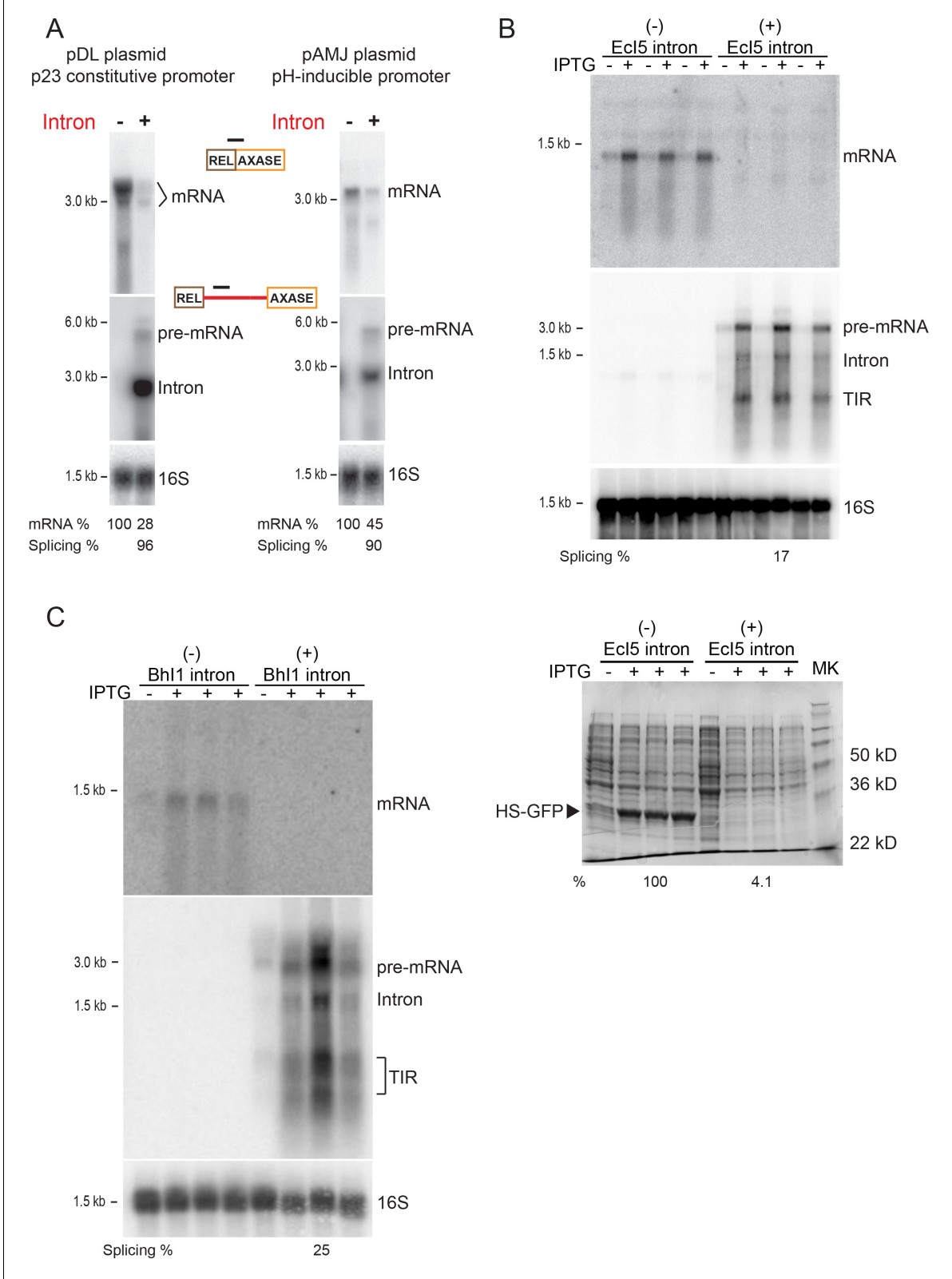

**Figure 2.** Reduction in gene expression persists with distinct plasmids, promoters, and group II introns. (**A**) RNA expression in Int⁻/Int⁺ cells under p23 promoter on the pDL278 plasmid (left) and under pH-inducible promoter on the pAMJ328 plasmid (right). RNAs in the Int⁻/Int⁺ cells were analyzed by Northern blotting using probes specific for mRNA (top), intron RNA (middle) and 16S rRNA (bottom, loading control). Quantitation of mRNA and splicing, normalized to 16S rRNA, is denoted at the bottom of image. Splicing efficiency is defined as the percentage of the spliced intron relative to

*Figure 2 continued on next page*

*Figure 2 continued*

the sum of pre-mRNA and spliced intron. (B) EcI5 and (C) BhI1 group II introns cause reduced expression of mRNA. Group II intron-containing (EcI5[+]/BhI1[+]) or the ligated exons (EcI5[-]/BhI1[-]) were fused to the coding sequence of GFP reporter. RNAs were analyzed by Northern blotting using probes specific for mRNA (top), intron (middle), and 16S (bottom, loading control). Splicing efficiency is defined as the percentage of the spliced intron relative to the sum of pre-mRNA and spliced intron. Splicing of both introns was at relatively low efficiency (EcI5: 17%; BhI1: 25%), which could be attributed to an inherent property of these introns in *E. coli*, or to improper IEP protein folding. For EcI5, analysis of the HS-GFP protein on a coomassie stained 12% SDS-polyacrylamide gel and relative protein levels are shown (B, bottom panel). TIR = Truncated Intron RNA. Biological replicates, n = 3. IPTG induced (+) and uninduced (-) Int[-]/Int[+] strains are shown.

DOI: https://doi.org/10.7554/eLife.34268.009

into a previous study (*Novikova et al., 2014*) suggesting that these two disparate mobile elements,

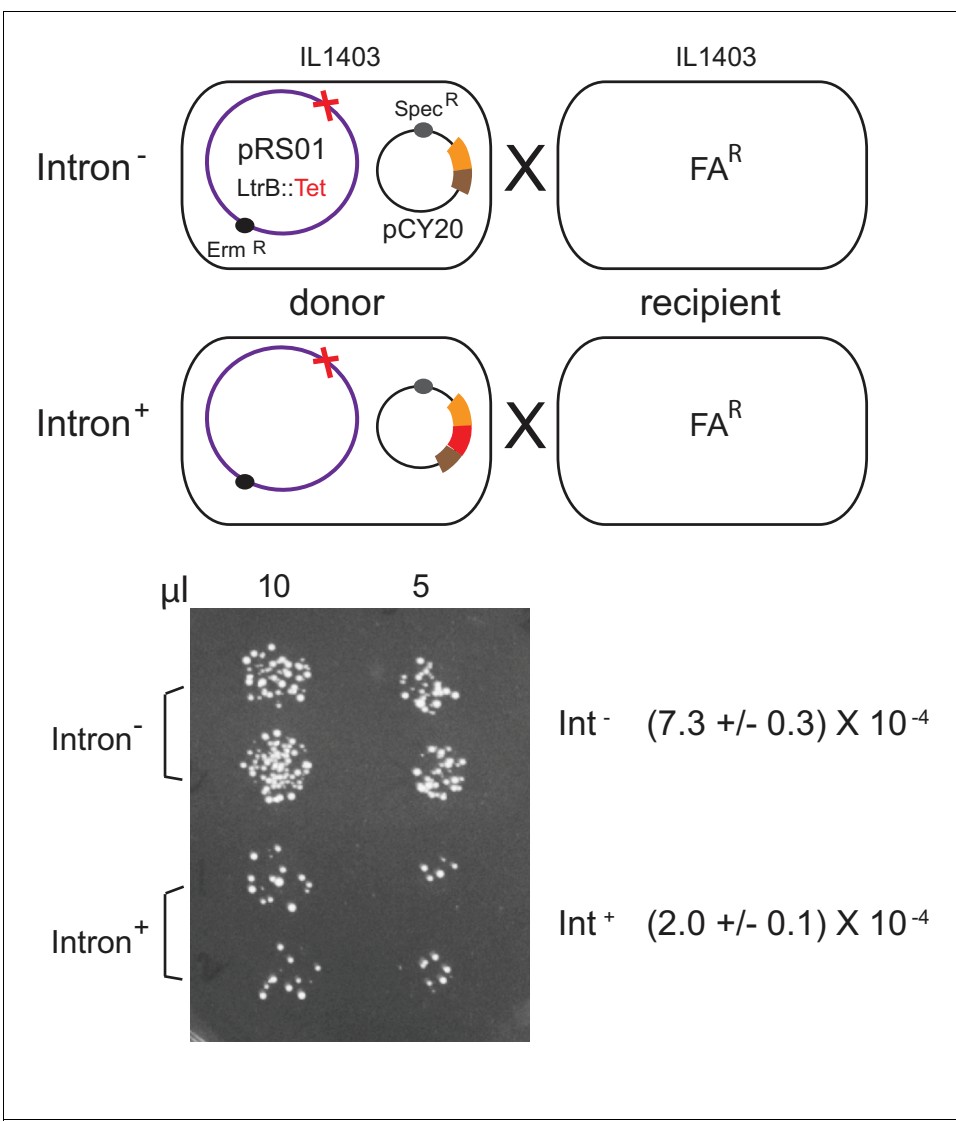

**Figure 3.** Inhibition of pRS01 conjugation by Ll.LtrB group II intron. Top, mating diagrams of IL1403 donor with IL1403-FA[R] recipient. Donor cells contain pRS01 (*erm*[R]), with *ltrB* interrupted by a *tet* gene (red cross), co-expressed with Int[-] or Int[+] (red bar) relaxase (brown and orange bars) from plasmid pCY20. Bottom, representative conjugation result. Spotting of 10 and 5 µl of donor plus recipient conjugation mix (two replicates per donor) on plates containing FA and Erm to select for transconjugants. Conjugation frequencies are shown (n = 2, p<0.05 based on two-tailed t-test with two-sample unequal variance).

DOI: https://doi.org/10.7554/eLife.34268.010

the mobile intron and the conjugative pRS01 plasmid, have a functional interplay.

## Spliced mRNA is associated with the intron

The first crystal structure of a group II intron lariat has revealed that ligated exons (mRNA) remain bound to the spliced intron RNA (*Robart et al., 2014*). The recently determined cryo-electron microscopy (cryo-EM) structure of the Ll.LtrB group II intron-IEP complex, that was isolated from its native host, also shows that after splicing the mRNA is retained within the ribonucleoprotein (RNP) (*Qu et al., 2016*). To confirm the presence of this intron-mRNA interaction in the bacterial cell, we performed intron-RNA pull-down assays (*Qu et al., 2014*). Briefly, a streptavidin-specific RNA aptamer was introduced into the Ll.LtrB intron in order to purify it over streptavidin resin (*Figure 4A*). After purification, the intron and any associated RNAs, were isolated and analyzed by reverse transcription. With the aptamer-containing intron, we observed that while the full-length pre-mRNA and spliced intron were pulled down as expected, the small mRNA (smRNA), generated from splicing of the aptamer-containing pre-mRNA with small exons (S-smRNA), was co-isolated. However, the smRNA produced either from the Int$^-$ cell (C-smRNA) or from the aptamer-free Int$^+$ cell (S-smRNA) were not co-isolated (*Figure 4B*). These results demonstrate that the intron is interacting with the spliced mRNA in the bacterial cell.

## Intron-mRNA interaction inhibits gene expression

Next, we investigated the possibility that the intron-mRNA interaction might account for the group II intron-promoted mRNA reduction. Similar to a previous study, we developed an in trans, two-plasmid expression system (*Qu et al., 2014*). We therefore co-expressed the intron-less *ltrB* relaxase from the pCY20 plasmid with either the Ll.LtrB intron-containing (with flanking small exons), or intron-less (small exons only) second plasmid, pLNRK (*Figure 5A*, left). Northern blotting analysis again showed that the level of relaxase mRNA dramatically dropped in the presence of the intron (21% vs. 100%). Additionally, an unexpected band smaller than the mRNA was detected (we call this band RNA 3, see below) (*Figure 5A*, right). Also, Western blotting indicated less accumulation of LtrB relaxase protein (24% vs. 100%).

It was necessary to elucidate if mRNA loss in this case is due to retrohoming or RNA-RNA interaction alone. To this end, mutants of the intron's IEP, LtrA, in which retrohoming was stopped due to inactivation of either the reverse transcriptase (RT) or DNA endonuclease (EN), were created (*Figure 5—figure supplement 1A*). RNA analysis showed that strong mRNA inhibition still persisted in these mutants (*Figure 5—figure supplement 1B*), thus indicating that the inhibition of gene expression was due to the proposed RNA-RNA interaction rather than retrohoming.

Using the *trans*-expression system, we validated that the intron-mRNA interaction requires EBS-IBS base-pairing, as shown previously in yeast (*Qu et al., 2014*). Mutants with nucleotide substitutions either in the IBS of the relaxase mRNA, or in the EBS in the intron expressed in trans, were compared to the respective wild-type counterparts (*Figure 5B,C*). The results showed that the relaxase mRNA level was recovered in the EBS mutant strain (*Figure 5C*, 21% to 92%), and increased ~2-fold in the IBS mutant (*Figure 5D*, 24% to 45%). Similar results were obtained with a different set of IBS-EBS mutants (m*), with recovery of mRNA levels in the EBS and IBS mutants from 23% to 116% and 23% to 51% respectively (*Figure 5—figure supplement 2*). Importantly, relaxase mRNA level was again reduced when the IBS mutation in the mRNA complemented the mutated EBS from the intron in the cell (*Figure 5—figure supplement 2*, down to 18%).

## The spliced intron retargets the mRNA

The observation that splicing ability of the intron is required for mRNA inhibition (data not shown) raised the possibility that the intron interaction with mRNA enables biochemical reactivity. We were also curious about the unexpected bands (RNAs 3 and 4) observed in *Figure 5C and D* respectively. Biochemical reactivity was validated and the origin of the extraneous bands was examined with the in trans system through a series of Northern blotting assays with multiple probes and confirmed by Rapid Amplification of cDNA Ends (RACE), which generates a cDNA copy for sequencing of the desired RNA. Four RNA molecules (RNA 1-RNA 4) with identity clearly distinct from the RNAs that were predicted to be expressed, were revealed (*Figure 6A–C*, *Figure 6—figure supplement 1A–D*, *Figure 6—figure supplement 2*).

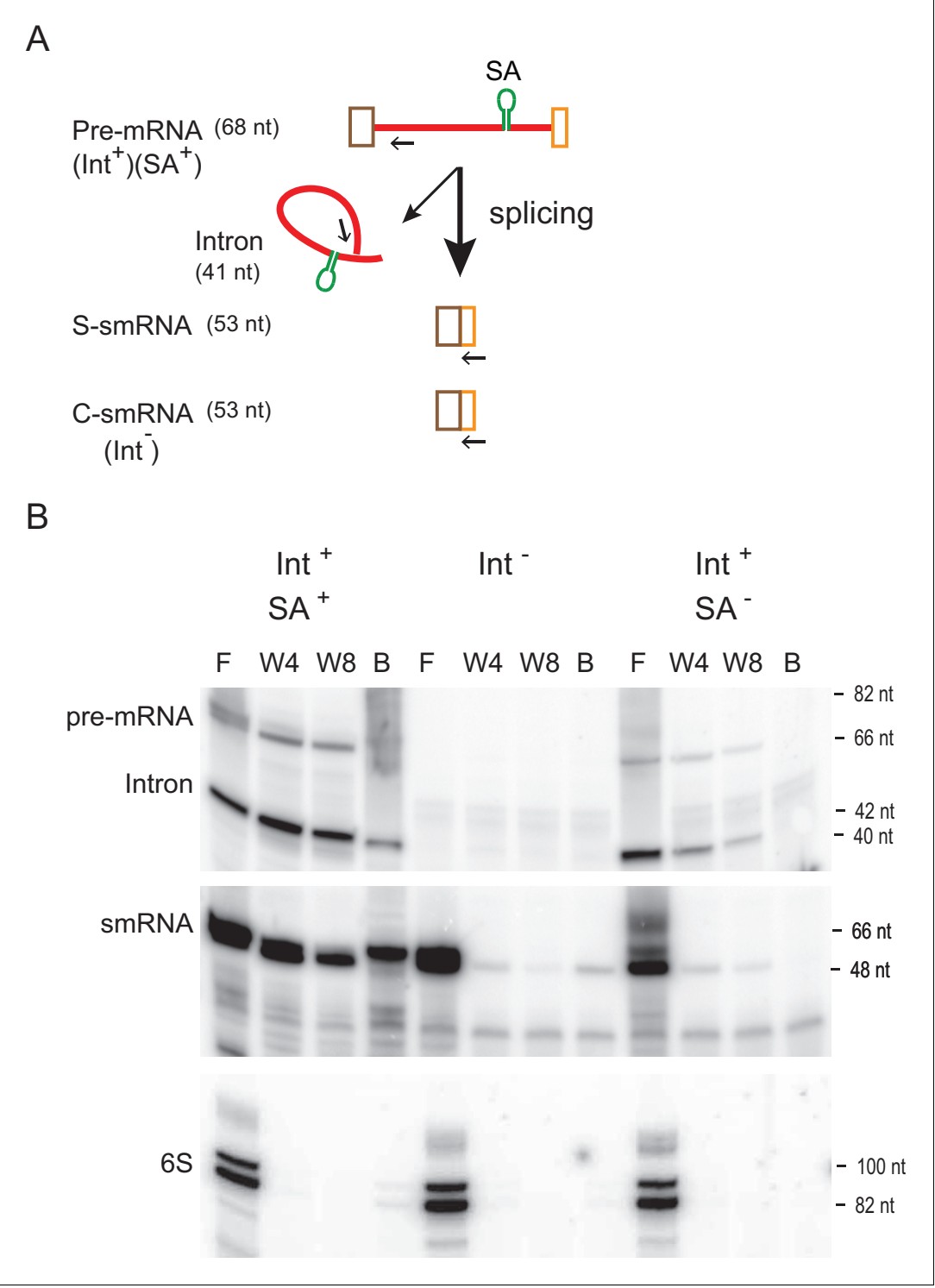

**Figure 4.** RNA-RNA interaction between intron and mRNA. (**A**) Schematic of construct harboring RNA aptamer. Pre-mRNA bears the Ll.LtrB intron (red line), which is flanked by small exons (brown and orange boxes). The RNA streptavidin aptamer (SA) is shown as a green stem-loop. Splicing of the aptamer-containing (Int+SA+) or aptamer-less (Int+SA-) pre-mRNA produces small ligated exons (S-smRNA) that have the same sequence as the control small mRNA (C-smRNA) generated from the intron-less construct (Int-). Primers IDT5078 and IDT1073 were used for analysis of smRNAs and intron RNAs (pre-mRNA and spliced intron), respectively, in panel below and are indicated as arrows. (**B**) mRNA binds to the intron (representative result of three biological replicates). RNAs with the SA aptamer were purified using streptavidin resin and were detected by reverse transcription using 5'- $^{32}$P-

*Figure 4 continued on next page*

*Figure 4 continued*

labeled primers specific for smRNAs and intron RNAs (pre-mRNA and Intron). The cDNA products were resolved on an 8% urea-polyacrylamide gel. Cellular 6S non-coding RNA was used as a loading control. F, flow-through; W4, the fourth wash; W8, the eighth wash, B, resin-bound.

DOI: https://doi.org/10.7554/eLife.34268.011

Among the four RNA molecules, two of them appeared in the intron-specific probing (*Figure 6A*; *Figure 6—figure supplement 1A*, lane 7). The larger of the two, termed RNA 1, was also revealed with both relaxase mRNA 5'- and 3'-exon probes (*Figure 6B–C*; *Figure 6—figure supplement 1C–D*, lane 7), and was identified as the intron-containing relaxase pre-mRNA. This was confirmed by probing a splicing-inactive catalytic triad mutant (T), which produced only this pre-mRNA band, RNA 1, and not RNAs 2–4 (*Figure 6—figure supplement 1A,C,D*, lane 3), and also by 5'-RACE followed by PCR amplification and sequencing of the cDNA (*Figure 6—figure supplement 2A*). This result suggests that the intron expressed in trans, is reverse splicing into the relaxase mRNA (*Figure 6A*). The smaller of the two intron-specific RNAs, termed RNA 2, was also revealed with the probe for 3'-exon but was absent in the probing for the 5'-exon (*Figure 6B–C*; *Figure 6—figure supplement 1C–D*, lane 7). With selective Northern blotting, along with 5'-RACE followed by cDNA sequencing, this RNA was identified as the unspliced intron expressed in trans, with its small 3'-exon replaced by the full-length 3'-exon of the relaxase mRNA (*Figure 6A–B*, *Figure 6—figure supplement 1A,D*, *Figure 6—figure supplement 2A*). This result suggests that there was shuffling of exons or RNA recombination, possibly between two reverse splicing reactions, that resulted in the formation of a chimeric precursor (*Figure 6B*).

The third RNA, RNA 3, previously seen in *Figure 5*, was observed in both the exon splice junction (*Figure 6—figure supplement 1B*, lane 7) and the relaxase mRNA 3'-exon-specific probings (*Figure 6B*; *Figure 6—figure supplement 1D*, lane 7), but it was absent from the probing for 5'-exon of the relaxase (*Figure 6C*; *Figure 6—figure supplement 1C*, lane 7). This RNA was further defined by 5'-RACE followed by cDNA sequencing as a ligation product of the small 5'-exon expressed in trans and the 3'-exon of the *ltrB* relaxase mRNA (*Figure 6—figure supplement 2A*). Identification of RNA 3 suggested that RNA 2, the chimeric intron precursor resulting from reverse splicing and RNA recombination described above, could be subject to forward splicing in the cell (*Figure 6B*). This result is in accord with splicing being favored over reverse splicing in vivo. Notably, RNA 3 appeared much more abundant than the relaxase mRNA (*Figure 6B*; *Figure 6—figure supplement 1B,D*).

Besides reverse splicing and its related RNA recombination, these experiments also suggested the occurrence of a **s**pliced **e**xons **r**eopening (SER) reaction (*Jarrell et al., 1988*) (*Figure 6C*). This was evidenced by identification of the fourth RNA product, RNA 4, a free 5'-exon of the relaxase mRNA. RNA 4 was observed in the probing specific for the 5'-exon of the relaxase mRNA (*Figure 6C*; *Figure 6—figure supplement 1C*, lane 7) but was absent with all other probes (*Figure 6A–B*; *Figure 6—figure supplement 1A,B,D*, lane 7). Additionally, its presence was validated by 3'-RACE analysis followed by PCR amplification and sequencing of the cDNA products (*Figure 6—figure supplement 2B*). However, free 3'-exon of the relaxase mRNA, which should be produced simultaneously with the free 5'-exon from the SER reaction, was not detected either by Northern blotting or 3'-RACE (*Figure 6B*; *Figure 6—figure supplement 1D*, lane 7; *Figure 6—figure supplement 2B*). This could be attributed to bacterial mRNAs usually having a triphosphate at the 5' end that is more resistant to mRNA decay than monophosphate (*Celesnik et al., 2007*; *Schoenberg, 2007*). Notably, this free 5'-exon RNA was also revealed when the intron was expressed in cis (*Figure 6C*; *Figure 6—figure supplement 1C*, lane 2; *Figure 6—figure supplement 2B*). Thus, these results suggest the SER, followed by RNA decay of the opened 3'-exon, to be a mechanism resulting in mRNA loss in the cell.

To further elucidate how these retargeting reactions are responsible for mRNA disappearance, we sought to examine mutants that were already created for this study. Initially we chose to identify the four RNA molecules in the intron_EBSm* and mRNA_IBSm* mutants (*Figure 5—figure supplement 2*). Using the in trans system, Northern blotting analyses showed that none of the four RNA products 1–4, that appeared with the wild-type strain existed in the mutants (*Figure 7*, lanes 3,5

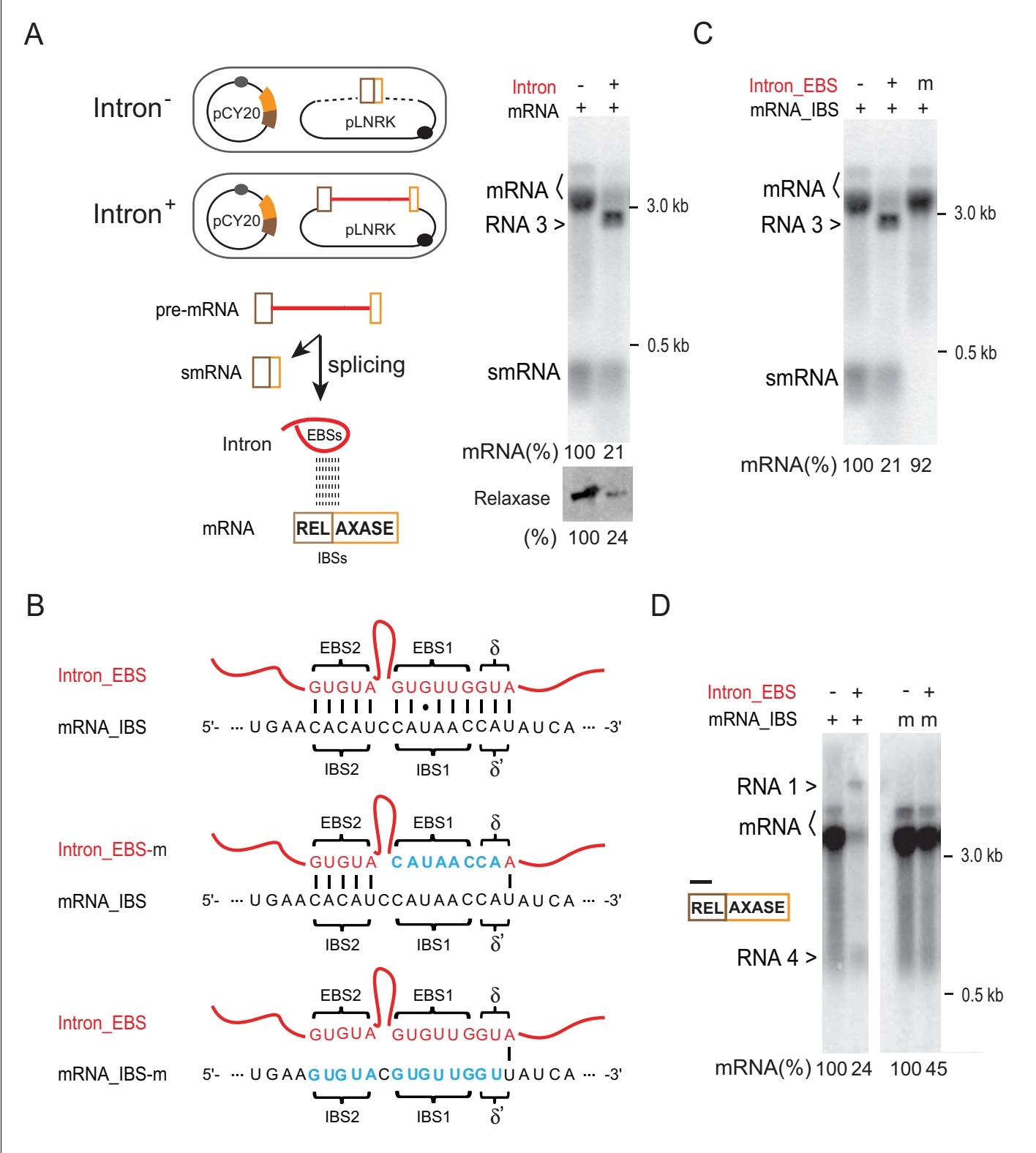

**Figure 5.** Intron-mRNA interaction causes mRNA inhibition. (**A**) Intron in trans expression system. In the schematic, *ltrB* relaxase mRNA (brown and orange bars), from plasmid pCY20, was co-expressed with the Ll.LtrB group II intron (red line) flanked with small exons (brown and orange boxes) from plasmid pLNRK (*cam^R*) (Intron+). Co-expression of the mRNA with an intron-less allele was used as the control (Intron-). This system was used in assays shown in panels C-D. Splicing of the intron-containing pre-mRNA generates small mRNA (smRNA) and Intron, and the latter is proposed to interact with the mRNA via EBS and IBS base pairing. To the right, mRNAs were analyzed by Northern blotting (top) as in *Figure 1B*, and LtrB relaxase protein

*Figure 5 continued on next page*

*Figure 5 continued*

was analyzed by Western blotting as in **Figure 1C** (bottom). Representative data of three biological replicates is shown. Quantitation of mRNA (two bands, bracketed) and protein levels are denoted at the bottom of the images. Dagger (>), unexpected product (identified in **Figures 5** and **6** as RNA 3). (B) EBS–IBS interaction sequences. WT (wt) and mutated (m) Intron_EBS (red) and mRNA_ IBS (black) sequences are shown with Watson–Crick pairs between them indicated as vertical bars and the wobble U:G pairs shown as a dot. Nucleotide substitutions in mutated EBS-IBS sequences are shown in blue. (C) Effect of EBS mutation on mRNA levels. mRNAs (two bands, bracketed) were analyzed by Northern blotting (top) as in **Figure 1B**. Quantitation of mRNA levels (derived from three biological replicates) is denoted at the bottom. Dagger (>), unexpected product (identified in **Figures 5** and **6** as RNA 3). (D) Effect of IBS mutation on mRNA levels. RNAs were analyzed by Northern blotting with a 5'-exon specific probe (black bar). Quantitation of mRNA (two bands, bracketed) levels (derived from three biological replicates) is denoted at the bottom. Daggers (>), unexpected products (identified in **Figures 5** and **6** as RNA 1 and RNA 4).
DOI: https://doi.org/10.7554/eLife.34268.012
The following figure supplements are available for figure 5:

**Figure supplement 1.** Reduction of gene expression is independent of retrohoming.
DOI: https://doi.org/10.7554/eLife.34268.013
**Figure supplement 2.** Complementation with EBS-IBS mutants shows mRNA targeting.
DOI: https://doi.org/10.7554/eLife.34268.014

versus lane 2). However, they reappeared when the EBS from the intron was mutated to complement the IBS within the relaxase mRNA in this mutant (**Figure 7**, lanes 6). This result confirmed that all of the identified mRNA retargeting reactions rely on the intron-mRNA interaction that is based on EBS-IBS base pairing.

## Discussion

### Group II introns are inherent inhibitors of gene expression

Inspired by the recently determined group II intron RNP structure and by the distribution features of group II introns in bacterial genomes, we performed a comparative expression study of intron-containing and intron-less variants of otherwise identical genes, to investigate the impact of group II introns on host gene expression. We discovered that group II introns reduce gene expression by robustly decreasing the spliced mRNA level in bacteria (**Figure 1**). This group II intron-induced mRNA disappearance appears independent of plasmids and promoters from which the genes are expressed (**Figure 2**), and is independent of stresses and changes of environmental conditions (data not shown). While this intron-induced reduction of mRNA levels was revealed mainly for the well-defined group IIA intron Ll.LtrB and its host *ltrB* relaxase gene in its native bacterial host *Lactococcus lactis*, it was also shown for two other types of bacterial group II introns, the group IIB EcI5 and group IIC BhI1 introns (**Figure 2**). Using the Ll.LtrB intron as the model, we also demonstrated that this mRNA reduction results from interaction between the intron with its spliced mRNA, via EBS-IBS base pairing (**Figures 4**, **5** and **7**). Since *ltrB* is part of an operon containing also *ltrC*, *ltrD*, *ltrE* and *ltrF* (**Chen et al., 2005**), it will be interesting to determine how *ltrB* targeting by the group II intron affects levels of the entire transcript.

Interestingly, group II introns can also cause gene silencing in eukaryotic cells (**Chalamcharla et al., 2010**; **Zerbato et al., 2013**). Here, EBS-IBS-based intron-mRNA interactions shut down nuclear gene expression (**Qu et al., 2014**). Taken together, we conclude that group II introns are inherent and general inhibitors of gene expression.

### How the mRNA level is controlled by the intron

By using the in trans system, we revealed that through RNA-RNA interactions, the spliced intron can retarget the mRNA in the cell (**Figures 6–7**, **Figure 6—figure supplement 1** and **2**). The intron's ribozyme activities include reverse splicing into the spliced mRNA, and the SER reaction. Because forward splicing is usually favored in the cell, based on the high splicing efficiency shown in **Figure 1B**, **Figure 1—figure supplements 1–2**, we speculate that reverse splicing accounts for a relatively small portion of mRNA loss. Additionally, a robust SER, which was also revealed in the in cis system (**Figure 6C**), splits the 5'-exon from the mRNA, and the free 3'-exon appears to be degraded (**Figure 8A**).

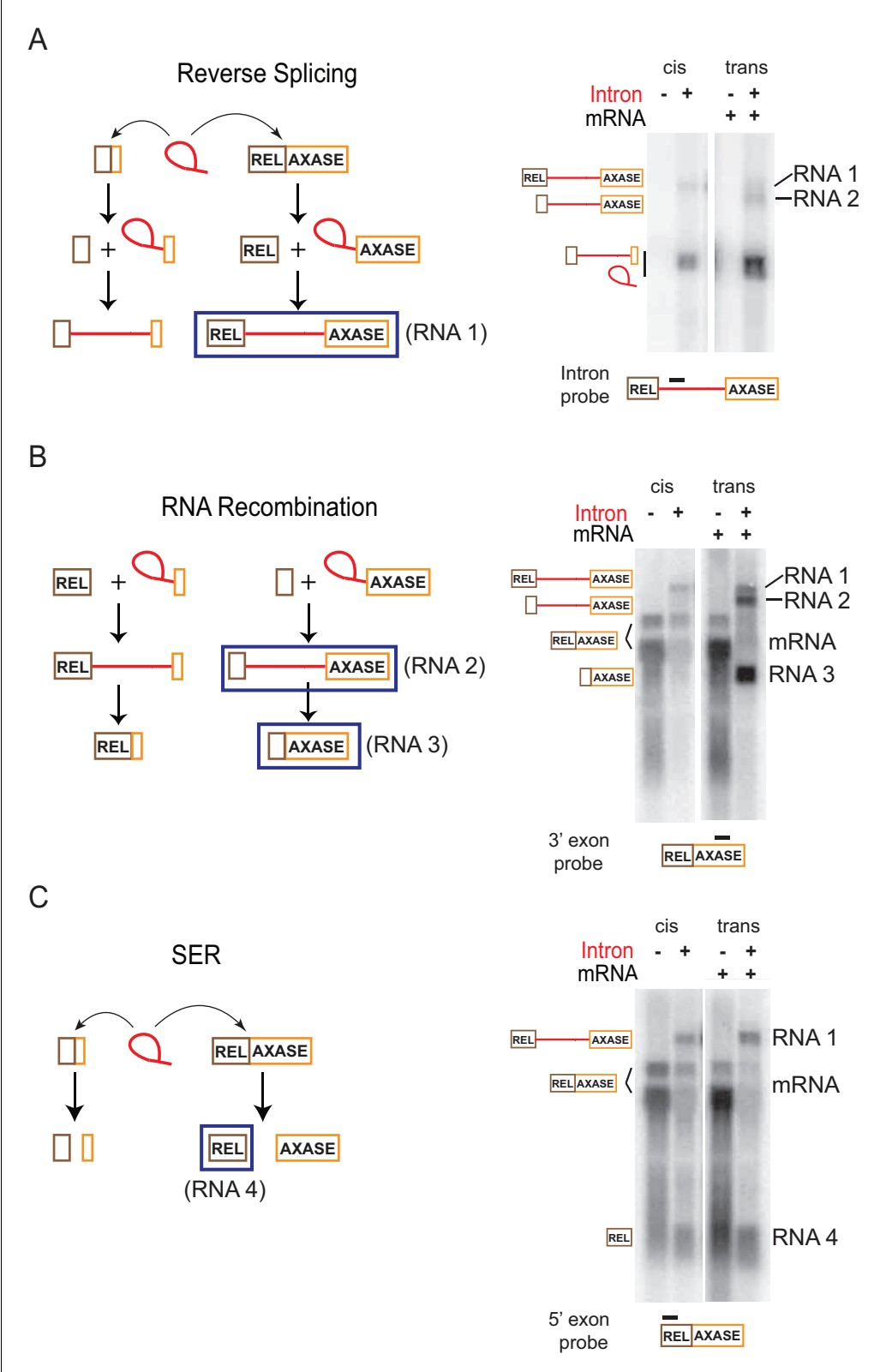

**Figure 6.** Retargeting mRNA by spliced intron. Proposed reactions in the in trans system as described in *Figure 5A*, are diagramed on the left and validated by Northern blotting (representative results of three biological replicates) on the right. (**A**) Reverse splicing into the mRNA. Left: The co-expressed full length *ltrB* relaxase mRNA or the smRNA (brown and orange boxes) are substrates of the intron ribozyme. Reverse splicing consists of two steps and generates two precursor forms. Right: RNAs expressed with the in trans system (right two lanes) were detected using an intron-specific

*Figure 6 continued on next page*

*Figure 6 continued*

probe (black bar), and compared to RNAs expressed in cis (left two lanes). This assay revealed both the predicted reverse splicing product (RNA 1), and an additional RNA product (RNA 2) resulting from the reaction shown in (B). (B) RNA recombination. Left: At the second step of reverse splicing, 5′-exons from the *ltrB* relaxase mRNA or the smRNA exchange with each other, thus producing chimeric intron precursors. These products could undergo forward splicing, thereby producing chimeric mRNAs. Right: RNAs expressed with the in trans system (right two lanes) were detected using a probe specific for the 3′-exon (black bar), and compared to RNAs expressed in cis (left two lanes). The mRNA was revealed as two bands (bracketed). This assay revealed both of the two predicted products resulting from RNA recombination (RNA 2, RNA 3), and also the RNA 1 product described in (A). (C) Spliced exons reopening (SER). Left: Either the relaxase mRNA or smRNA could be split by the intron at the exon-exon junction, thus freeing the 5′- and 3′-exons. Right: RNAs expressed with the in trans system (right two lanes) were detected using the probe specific for the 5′-exon (black bar), and compared to RNAs expressed in cis (left two lanes). The mRNA was revealed as two bands (bracketed). This assay revealed the predicted free 5′-exon of the relaxase mRNA (RNA 4) resulting from SER in both the in trans and in cis systems, and also the RNA 1 product described in (A).

DOI: https://doi.org/10.7554/eLife.34268.015

The following figure supplements are available for figure 6:

**Figure supplement 1.** Characterization of mRNA retargeting reactions.
DOI: https://doi.org/10.7554/eLife.34268.016

**Figure supplement 2.** Identification of mRNA targeting products 1–4 by 5′ and 3′ RACE.
DOI: https://doi.org/10.7554/eLife.34268.017

Although group II introns commonly invade DNA sequences, targeting RNA in vivo has not been previously described. However, RNP invasion of DNA is a relatively unfavorable event (*Aizawa et al., 2003*), which makes it tempting to speculate that the RNA SER and subsequent RNA degradation may favor DNA invasion and retromobility.

The RNA decay study with rifampicin indicated slower mRNA degradation in the Int$^+$ cell (*Figure 1—figure supplement 3*). However, this result may be misleading as the residual RNA that escapes intron targeting may be a more stable fraction. Alternatively, the intron may act as ribosomes do on bacterial mRNA, providing a 'protective barrier' by binding the mRNA that escapes degradative attack by the group II intron (*Deana and Belasco, 2005*). Additionally, enhanced stability may be a reflection of the reciprocal relationship between the stability and concentration of bacterial mRNAs (*Nouaille et al., 2017*), such that lower mRNA levels in the Int$^+$ cell, would lead to higher stability of the mRNA. Whichever mechanism underlies the phenomenon, the consequence is that a fraction of the mRNA is maintained for translation.

## Implications for distribution and spread of group II introns

The discovery that group II introns inhibit gene expression likely explains why they are located mostly outside of genes or in non-essential genes, and why they are maintained at low frequency in bacterial genomes (*Dai and Zimmerly, 2002a*; *Zimmerly and Semper, 2015*). Group II intron-promoted reduction of gene expression could be the major driver that selected for group II introns residing in intergenic or non-essential regions. It is therefore intriguing that there are rare cases of essential genes containing group II introns in bacterial genomes (*Candales et al., 2012*; *Dai et al., 2003*; *Dai and Zimmerly, 2002a*; *Zimmerly and Semper, 2015*). We speculate that, in these cases, either robust gene expression is not needed, that reduction in gene expression is modest or that the intron plays a regulatory role (*Belfort, 2017*). The effect of organellar group II introns that are commonly present in essential genes is largely unknown (*Dai and Zimmerly, 2002a*; *Zimmerly and Semper, 2015*). Perhaps many of these introns have lost the ability to target mRNA and cause its loss, or do so very inefficiently. Alternatively, a lower expression level might be required for proper functions of these host genes. Notably, favoring the latter notion, a recent study showed that the presence of self-splicing introns including group II introns in the mitochondrion of *Saccharomyces cerevisiae* was required for inefficient expression of host genes that was essential for maintaining proper mitochondrial function (*Rudan et al., 2018*).

Group II introns are also frequently found in mobile DNAs (*Candales et al., 2012*; *Dai et al., 2003*; *Dai and Zimmerly, 2002a*; *Zimmerly and Semper, 2015*). One of the examples is the Ll.LtrB group II intron that naturally resides in the conjugative plasmid pRS01, interrupting the *ltrB* relaxase gene whose expression is required for the horizontal gene transfer (HGT) of the plasmid. A recent study indicated that pRS01 promotes retromobility of the intron by providing LtrB relaxase that stimulates both the frequency and diversity of retrotransposition (RTP) events with its off-target DNA

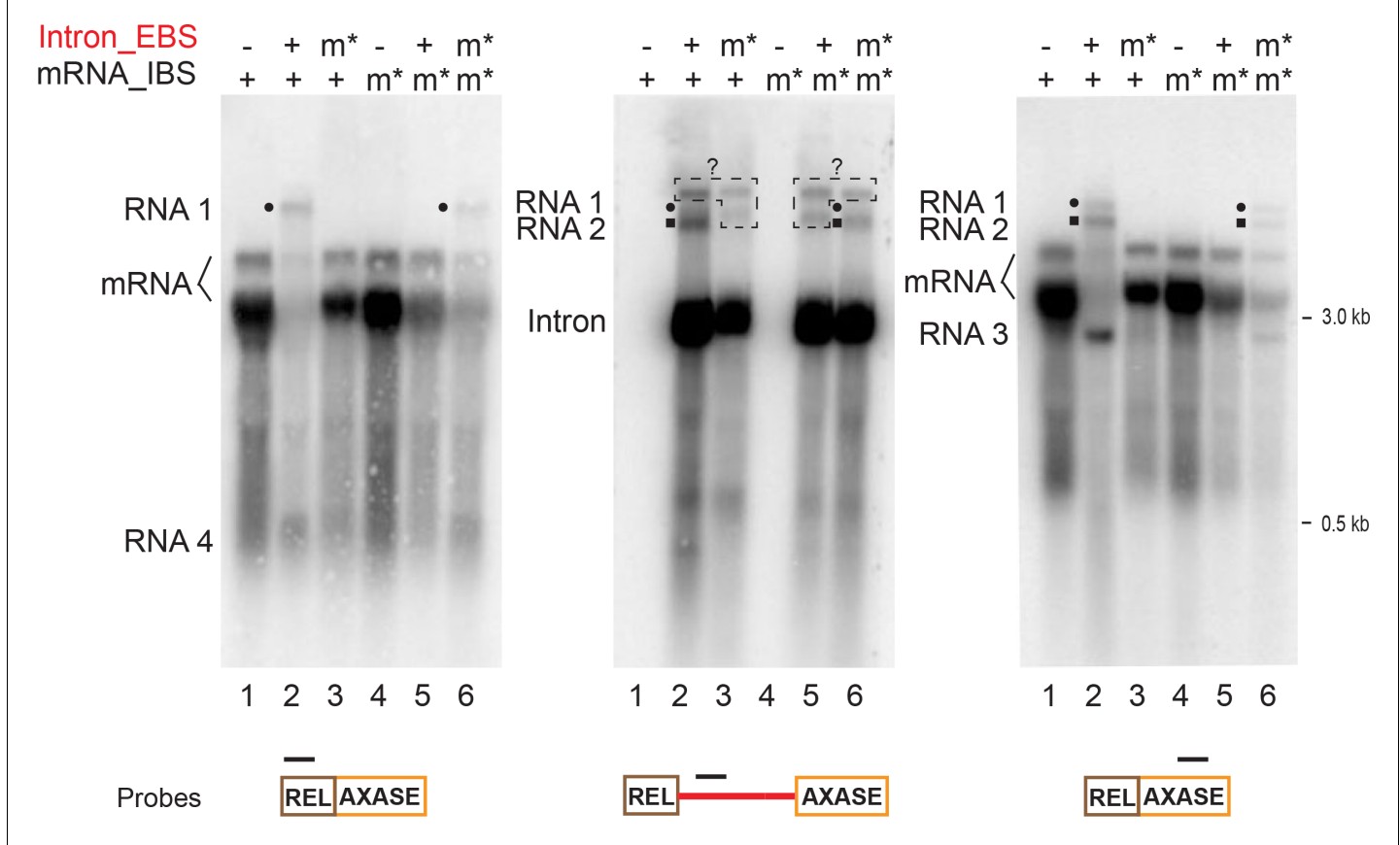

**Figure 7.** mRNA retargeting requires EBS-IBS interaction. RNAs expressed using the in trans system (*Figure 5A*), with both wild-type mRNA (lanes 1–3) or the mRNA_IBS mutants (m*, lanes 4–6), in the presence or absence of the wild-type intron (+/-) or the intron_EBS mutant (m*), were analyzed by Northern blotting as in *Figure 6* and *Figure 6—figure supplement 1* and representative results of three biological replicates are shown. The mRNA retargeting products (RNAs 1–4, and RNAs 1 and 2 denoted with a black circle or square, respectively) indicated on the left, appear with the wild-type mRNA (two bands, bracketed) in the presence of the wild-type intron. RNA signals revealed by intron-specific probing (middle panel) are boxed and denoted with '?' as their identities are unknown, because they were absent in the 5'- and 3'-exon specific probing (left and right panels). RNAs 1–4 are absent in the mutants Intron_EBSm* (lanes 3) and mRNA_IBSm* (lanes 5) where EBS-IBS pairing cannot form. Notably, these bands appear when the two mutants can complement each other by EBS-IBS pairing (lanes 6).

DOI: https://doi.org/10.7554/eLife.34268.018

nicking activity (*Novikova et al., 2014*). Here our study showed that the Ll.LtrB group II intron reduces the conjugal transfer of pRS01 via inhibition of its host gene expression (*Figure 3*). Thus, these findings together suggest that the two mobile genetic elements (MGEs) functionally interact by exploiting *ltrB* relaxase gene expression. Whereas relaxase expression stimulates intron RTP and HGT of the conjugative element, inhibition of relaxase expression by the group II intron opposes the promotion of RTP and HGT (*Figure 8B*), effects that are likely in equilibrium. In general, expression of conjugative transfer genes is tightly controlled to minimize the burden on the host (*Zatyka and Thomas, 1998*). Indeed conjugative plasmid gene expression is kept in a default 'off' state and is switched on only under conditions that are optimal for transfer of the conjugative element (*Singh and Meijer, 2014*). Silencing of the relaxase gene by its resident group II intron therefore represents a novel way in which a conjugative element is down-regulated.

## Evolutionary implications

The demonstration that intron RNAs are not only able to transpose and cleave exogenous RNAs, but can also recombine RNAs from different sources, suggests that progenitors of group II introns could have processed RNAs in primitive genomes. These actions could have contributed to the generation of an evolved genome and a cell that has novel functions and evolutionary advantages.

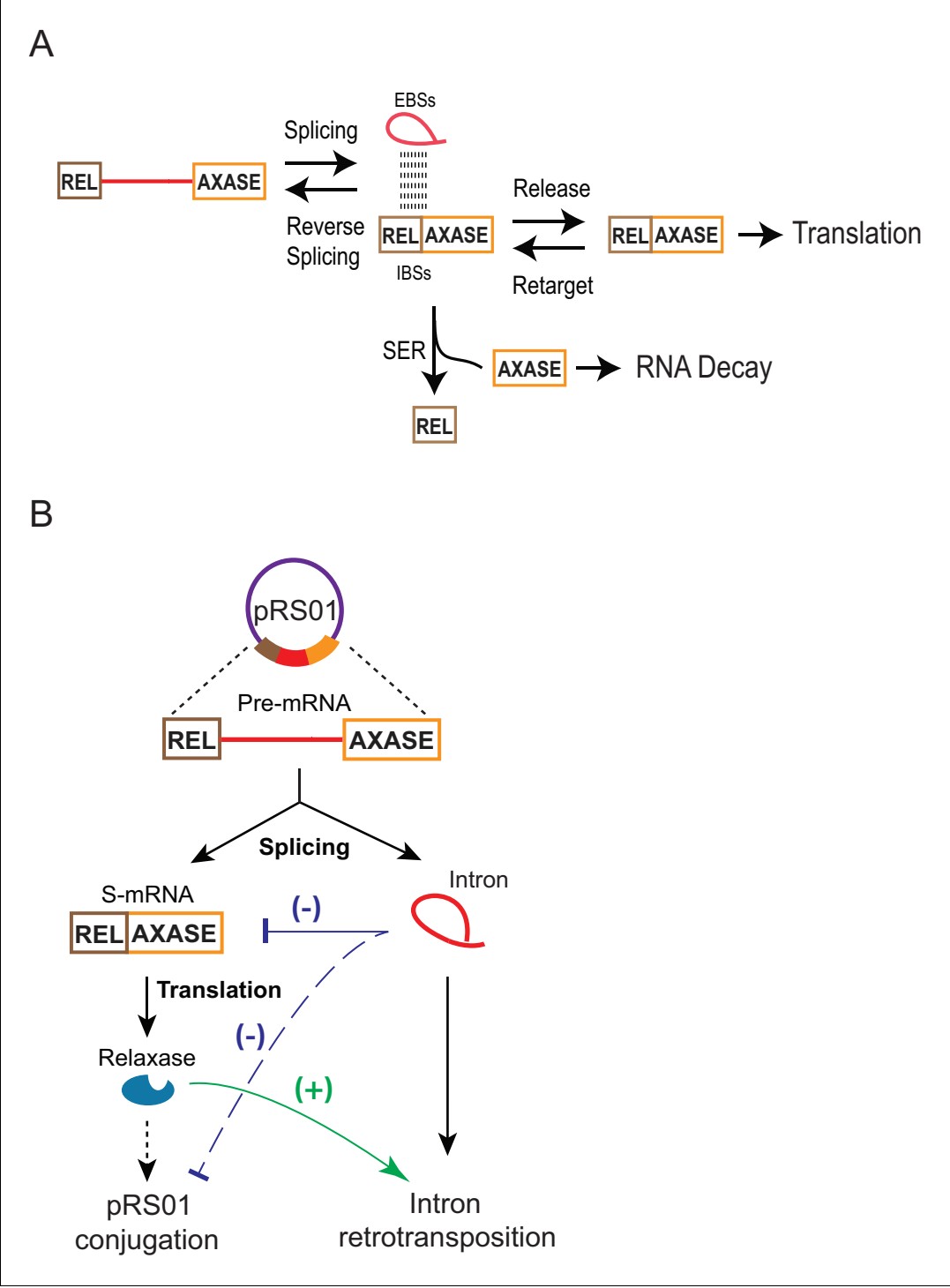

**Figure 8.** Model for group II intron regulation of gene expression and pRS01 conjugation. (**A**) How the group II intron controls the mRNA. Intron-mRNA interaction via EBS-IBS base pairing is the basis of three biochemical processes: splicing, reverse splicing, and spliced exons reopening (SER). While all these reactions could affect mRNA levels, SER is a major driver of the mRNA loss. It splits the mRNA into free 5'- and 3'-exons, the latter of which is susceptible to cellular RNA decay. Meanwhile, the intron could function as a 'shelter' that protects the mRNA that survives SER and is being retained within the intron from RNA decay. Because of these contradictory effects and equilibrium between pairs of reversible processes, a fraction of mRNA can be maintained for translation. (**B**) Functional interactions of the Ll.LtrB group II intron with the conjugative plasmid pRS01 through the intron host gene *ltrB* relaxase. Stimulation of retrotransposition by relaxase (green line), has been previously
*Figure 8 continued on next page*

*Figure 8 continued*

reported (*Novikova et al., 2014*). Reduced expression of *ltrB* relaxase by the intron (blue line) and its resulting reduction in pRS01conjugtaion (dashed blue line) as revealed in this study are shown.

DOI: https://doi.org/10.7554/eLife.34268.019

In regard to the ancestral relationship between group II and spliceosomal introns, expression of group II introns in nuclei causes gene silencing in eukaryotic cells (*Chalamcharla et al., 2010*; *Zerbato et al., 2013*). In this study, we demonstrated that group II introns also hinder gene expression in bacteria. Notably, a common mechanism underlying the group II intron-promoted inhibition of gene expression in bacteria and eukaryotes is based on interactions between intron and mRNA via EBS-IBS base pairing (*Qu et al., 2014*). In contrast to group II introns, many spliceosomal introns are shown to enhance gene expression in plants and other organisms (*Le Hir et al., 2003*; *Rose, 2008*). Interactions between the intron and mRNA no longer exist in the splicing of spliceosomal introns, where such pairings have been functionally substituted with interactions between U5 snRNA and mRNA (*Hang et al., 2015*; *Qu et al., 2016*; *Yan et al., 2015*). We speculate that, through these changes, introns can promote mRNA levels and gene expression, or at very least not be inhibitory, and therefore provided a force that could have stimulated evolution of group II introns into spliceosomal introns in primitive nuclear genomes.

# Materials and methods

**Key resources table**

| Reagent type (species) or resource | Designation | Source or reference | Identifiers | Additional information |
|---|---|---|---|---|
| Strain, strain background (*Lactococcus lactis*) | IL1403 | PMID: 11337471 | | |
| Strain, strain background (*Lactococcus lactis*) | IL1403 (FA-R) | Belfort Lab | | Fusidic Acid Resistant |
| Strain, strain background (*Escherichia coli*) | MC1061 | PMID:6997493 | | |
| Antibody | LtrB antibody | Gary Dunny, personal communication | | |
| Commercial assay or kit | QuikChange Lightning Site-Directed Mutagenesis Kit | Agilent, Santa Clara, CA | 210519 | |
| Commercial assay or kit | iScript cDNA Synthesis Kit | Bio-Rad, Hercules, CA | 1708890 | |
| Commercial assay or kit | iTaq Universal SYBR Green Supermix | Bio-Rad | 172–5120 | |
| Commercial assay or kit | 5' RACE System | Invitrogen, Carlsbad, CA | 18374058 | |
| Commercial assay or kit | 3' RACE System | Invitrogen | 18373019 | |
| Commercial assay or kit | SMARTer RACE 5'/3' kit | Takara, Mountain View, CA | 634860 | |

## Strains and growth conditions

*Lactococcus lactis* IL1403 was used for *ltrB* gene expression and RNA and protein analysis. IL1403 pRS01 *ltrB*⁻ (ΔLtrB::tet) and IL1403 (FA$^R$) were used as donor and recipient strains for conjugation, respectively. Cultures were grown in M17 media supplemented with 0.5% glucose (GM17) in tightly-capped tubes or bottles at 30°C without shaking. For gene expression, cultures were grown to OD$_{600}$ ~0.6 and nisin was added to a final concentration of 0.4 µg/ml, for 2–3 hr. Cultures were spun at 5000 x g and pellets were stored at −80°C. Where appropriate, the media contained spectinomycin (Spec) at 300 µg/ml, chloramphenicol (Cam) at 10 µg/ml, erythromycin (Erm) at 10 µg/ml, or fusidic acid (FA) at 25 µg/ml.

*Escherichia coli* MC1061(DE3) was used for over-expression of Ecl5 and BhI1 plasmids. Cultures were grown in LB media with 100 µg/ml ampicillin at 37°C with aeration. At OD$_{600}$ ~0.3, cultures

were induced with 0.1 mM IPTG for 3 hr. Cultures were spun at 5000 x g and pellets were stored at −80°C.

## Plasmids

All plasmids created or used are listed in *Supplementary file 1*, and DNA oligonucleotides used in this study are listed in *Supplementary file 1*. Plasmids pCY20LtrB (Int⁻) and pCY20LtrB::Ll.LtrB (Int⁺) were the vector pair used in the intron in cis system for comparative analysis of *ltrB* relaxase gene expression in the absence and presence of the Ll.LtrB intron. With the in trans system, pCY20LtrB was used for *ltrB* relaxase gene expression while plasmid pLNRK smEx::Ll.LtrB was used to express the intron. For an intron-less control in this system, pLNRK smEx::Ll.LtrB was replaced with plasmid pLNRK smEx, that contained only the small flanking exons. The *ltrB* IBS mutants, mRNA_IBSm and mRNA_IBSm*, and Ll.LtrB intron mutants including EBSm, EBSm* and Triad, and the IEP EN and RT mutants were created with the above pCY20 and pLNRK plasmids by site-directed mutagenesis (SDM) using QuickChange Lightning kit (Agilent, Santa Clara, CA). More specifically, to create EBSm*, the IBS region of the small exons of EBSm were mutated to make the intron splicing competent. This was done with 2 rounds of SDM PCR, using the plasmid product from PCR1 as the template for PCR2. All plasmids were confirmed by sequencing (see oligonucleotides used in *Supplementary file 1*). Construction of other plasmids used in this study are detailed below.

### pDL278 and pAMJ328 expression plasmids

For cloning of pDL278 (*LeBlanc et al., 1992*) and pAMJ328 (*Jørgensen et al., 2014*) constructs, *ltrB* -/+Ll.LtrB intron inserts were PCR amplified from pCY20 plasmids using primers with SphI and BamHI sites (IDT4767 and 4768, respectively) for pDL278, or primers with SpeI and PstI sites (IDT4766 and 4769, respectively) for pAMJ328. The PCR products were first cloned into pGEM-T (Promega, Fitchburg, WI) as per manufacturer's protocol and then cut and pasted into respective parental plasmids.

### EcI5 and BhI1 expression plasmids

HS-GFP cloning: Homing site (HS) sequence of EcI5 or BhI1 was fused with GFP coding sequence directly by PCR of plasmid pFA6a-GFP(S65T)-KanMx6 (Addgene, Cambridge, MA) using primer pairs IDT4831/4827 and IDT4826/4827 respectively, and with HindIII and SpeI restriction sites inserted respectively in the 5′- and 3′-termini of the amplicons. GPII-GFP cloning: EcI5 or BhI1 group II intron full-length sequences was amplified by PCR of genomic DNAs provided by Dr. Alan Lambowitz using primer pairs IDT4832/4833 and IDT4828/4829 respectively, and with HindIII restriction site inserted in the 5′-terminus of the amplicons. GFP coding sequence was amplified by PCR of plasmid pFA6a-GFP(S65T)-KanMx6 (Addgene) using primer pairs IDT4834/4827 or IDT4830/4827 for SOEing of GFP with ECI5 or BhI1 introns. The GFP amplicon was then mixed and ligated with the respective EcI5 or BhI1 amplicons by PCR using primer pairs IDT4832/4827 and IDT4828/4827 respectively, and with HindIII and SpeI restriction sites inserted respectively in the 5′-and 3′-termini of the amplicons. pET11a plasmid was digested with NdeI and BamHI and ligated with pre-annealed DNA oligonucleotides IDT4824 and IDT4825 that contain HindIII and SpeI sites. The plasmid was digested with HindIII and SpeI, and ligated with digested EcI5 or BhI1 HS-GFP or GPII-GFP resulting in the plasmids pET11a-HS-GFP and pET11a-GPII-GFP.

### Plasmids for RNA pull-down

Streptavidin aptamer-containing Ll.LtrB intron was amplified from plasmid pGpII(SA)-CUP1(6XMS2) by PCR using primer pairs IDT4942/4943, digested with XhoI and NotI and ligated into digested plasmid pLNRK smEx-nLIC (*Qu et al., 2014*). The resulting plasmid, pLNRK smEx::Ll.LtrB ΔORF (SA)-nLIC, was used for expression of SA aptamer-containing intron. The same plasmid backbone was ligated with pre-annealed DNA oligonucleotides IDT5051 and IDT5052 that contain XhoI and NotI sites, resulting in the control plasmid, pLNRK smEx::Ll.LtrB ΔORF-nLIC.

## Northern blotting

For RNA analysis, 7.5 µg of total RNA was separated on a 1.2% agarose/formaldehyde gel and transferred to hybond XL membrane (GE Healthcare, Pittsburgh, PA), which was then UV cross-linked and

probed with $^{32}$P-labeled oligonucleotides using Rapid-hyb buffer (GE Healthcare). For the Ll.LtrB intron, the membranes were hybridized for detection of mRNA exon splice junction (IDT4685), 5′-Exon (IDT5374), 3′-Exon (IDT5012), and Ll.LtrB intron (IDT1073). For the EcI5 intron, membranes were hybridized for the detection of mRNA (IDT4972) and intron (IDT4970), and for BhI1 intron, membranes were hybridized for mRNA (IDT4975) and intron (IDT4973). All membranes were probed for 16S rRNA (IDT861) as a loading control. Images were exposed on a phosphor screen, scanned on a Typhoon Trio, and quantified using ImageQuant.

## Western blotting

For LtrB relaxase analysis, total cell lysate was separated on a 12% SDS-polyacrylamide gel and transferred to 0.2 µM Immuno-blot PVDF membrane (Bio-Rad) at 25V for 30 min. The membrane was blocked with 5% dry milk in TBS-T (20 mM Tris, 140 mM NaCl, 2% Tween), incubated with a 1/1500 dilution of primary anti-relaxase antibody for 1 hr, washed with TBS-T twice for 15 min, and incubated with a 1/10,000 dilution of secondary HRP-labeled anti-rabbit antibody (Advansta, Menlo Park, CA) for 1 hr. Chemiluminescent HRP substrate (Advansta WesternBright ECL) was used for detection. For lane normalization, total protein was visualized from a coomassie stained 12% SDS-polyacrylamide gel. All images were scanned using a Bio-Rad ChemiDoc MP. Relaxase bands and total protein were quantified using Bio-Rad Image Lab software.

## qRT-PCR

cDNA synthesis was performed in a 20 µl reaction with 10 ng of DNase treated (Promega RQ1 DNase) RNA template using iScript (Bio-Rad, Hercules, CA), as per manufacturer's protocol. Minus RT (RT$^-$) controls were also performed. Total RNA quality and primer specificity were analyzed by gel electrophoresis prior to qPCR. qPCR reactions were done in a total volume of 10 µl, using iTaq Universal SYBR Green Supermix (Bio-Rad) and contained 2 µl of the cDNA reaction as template and 5 pmol of each primer. Reactions were run in technical triplicates and a no-template control was included in every run. Reactions were amplified with the following conditions: 95℃ for 30 s, (95℃ for 5 s, 60℃ for 10 s) x 40 cycles, melt curve 65℃ to 95℃ (0.5℃ every 2 s), on a Bio-Rad CFX384 Touch Real-Time PCR Detection System. Three biological replicates were run for each sample. Amplification efficiencies (E) of all primers were calculated using a 10-fold dilution and standard curve. Efficiencies, $C_t$ values and charts were obtained using Bio-Rad CFX Manager Software. Primers used, their percent amplification efficiencies and amplicon lengths are listed in *Supplementary file 1*. Relative gene expression of each target (tar) was normalized to a reference gene (ref), CopA, whose expression was demonstrated to be constant in a previous study (*Magnani et al., 2008*), and calculated using the following equation, with subtraction of RT$^-$ background:

$$[(E_{tar})^{-Ct(mean,RT+)} - (E_{tar})^{-Ct(mean,RT-)}]/[(E_{ref})^{-Ct(mean,RT+)} - (E_{ref})^{-Ct(mean,RT-)}].$$

## Reverse transcription primer extension

To identify splicing products, primer extension was performed using SuperScript III Reverse Transcriptase (Thermo Fisher Scientific, Waltham, MA), as per manufacturer's protocol, with 4 µg of DNase-treated RNA and 0.4 pmol of $^{32}$P-labeled oligonucleotides. Oligonucleotide IDT4836 was used with the addition of ddTTP for detection of intron precursor and spliced intron. Products were separated on an 8% Urea-polyacrylamide sequencing gel. To measure 5′-end transcription levels of the mRNA and intron precursor, IDT4916 was used, and products were separated on a 10% Urea-polyacrylamide gel. To analyze RNAs that were pulled down with streptavidin resin, IDT5078, IDT1073 and IDT5127 were used to probe the smRNAs, intron RNAs, and 6S non-coding RNA, respectively. To measure splicing efficiency of the intron_EBSm* mutant, oligonucleotide IDT1073 was used to detect the presence of intron precursor and spliced intron, and products were separated on a 10% Urea-polyacrylamide gel. Images were exposed on a phosphor screen, scanned on a Typhoon Trio, and quantified using ImageQuant.

## mRNA degradation

*L. lactis* IL1403 was grown to OD$_{600}$ ~0.6 and nisin was added to a final concentration of 0.4 µg/ml for 0.5 hr. Rifampicin was then added to a final concentration of 0.2 mg/ml, and 10 ml of culture was

removed for each aliquot and immediately spun down at 4°C. Total RNA was prepared and analyzed for Northern blots.

## Polysome profiling

To perform polysome profiling, 200 ml of *L. lactis* IL1403 Int$^+$/Int$^-$ cells (1:20 dilution of saturated overnight culture) were grown and induced for gene expression with nisin for 2–3 hr. 100 mg/ml chloramphenicol was added and the cultures were chilled on ice for ~30 min with intermittent whirling and then collected by centrifugation at 5,000 rpm for 10 min at 4°C. Cell pellets were resuspended in 500 µl of ice-cold lysis buffer (20 mM Tris-HCl, pH 8.0; 140 mM KCl; 40 mM MgCl$_2$; 0.5 mM DTT; 100 µg/ml chloramphenicol; 1 mg/ml heparin; 20 mM EGTA; 1% Triton X-100) and washed twice with the same buffer. The cell pellets were resuspended again in 500 µl of lysis buffer and snap-frozen in liquid nitrogen. Then the cells were disrupted at 4°C in 15 ml falcon tubes with 500 µl of 0.1 mm ice-cold glass beads by rigorous vortexing (30 s for 20 times, with 1 min interval) and then briefly spun down at 4,000 rpm for 5 min. Crude cell lysate was then gently mixed, transferred into 1.5 ml tubes on ice, and cleared by 14,000 rpm for 25 min at 4°C. 400 µl of the cleared lysates were then loaded onto prepared 10–50% sucrose gradients in lysis buffer without 0.5 mg/ml heparin and centrifuged at 36,000 rpm for 153 min at 4°C (SW41 rotor). 30 fractions of ~400 µl each were collected for each gradient from top to bottom. RNAs were extracted from each fraction by using phenol/chloroform and analyzed by running gels and performing Northern blotting.

## Conjugation assays

To measure conjugation efficiencies, *L. lactis* IL1403 (FA$^R$) was used as the recipient strain and was grown to an OD$_{600}$ ~0.6 and then for an additional 3 hr. *L. lactis* IL1403 ΔLtrB::tet (*erm$^R$*) was used as the donor strain, and contained either the Int$^-$ plasmid pCY20LtrB, or the Int$^+$ plasmid pCY20LtrB::Ll. LtrB. Cultures were grown to OD$_{600}$ ~0.6 and nisin was added to a final concentration of 0.4 µg/ml for 3 hr. Equal volume of donor and recipient were mixed, spotted on pre-incubated filters (0.45 µm pore size, Millipore) on GM17 plates, and incubated ~18 hr at 30°C. The filters were removed from the plates, washed with 5 ml GM17 in a 50 ml conical tube by vortexing, and the wash was spotted or plated on GM17 plus 25 µg/ml fusidic acid and 10 µg/ml erythromycin for selection of transconjugants. Plates were incubated ~18 hr at 30°C. To calculate input donor, the donor strain was diluted to $10^{-6}$ and spotted or plated on GM17 plus 10 µg/ml erythromycin. Conjugation frequency was expressed as exconjugants per donor (Erm$^R$FA$^R$/Erm$^R$).

## RNA Pull-down

Bacterial cells (100 ml) were collected after 2–3 hr of nisin induction of RNA expression and disrupted in 800 µl of CB$_{500}$ solution (20 mM Tris–HCl, pH 8.0, 500 mM NaCl, 0.1 mM EDTA, 1 mM PMSF) by vortexing (30 s, rest for 1 min, 24 cycles) using 500 µl of 0.1 mm ice-cold glass beads (Sigma). Lysates were cleared by centrifugation at 14,000 rpm for 25 min. To pull down RNAs, ~500 µl of cleared lysate was incubated with 100 µl of streptavidin resin (Thermo Scientific) that was equilibrated with CB$_{500}$. The resin was washed eight times with 1 ml CB$_{500}$ buffer. RNAs bound to resin were eluted with 400 µl of 5 mM biotin for 1 hr and were extracted from the eluates. RNA identities were determined by using reverse transcription primer extensions.

## 5′ and 3′ RACE

To further identify mRNA targeting products 1–4, 5′ and 3′ RACE (<u>R</u>apid <u>A</u>mplification of <u>c</u>DNA <u>E</u>nds) experiments were performed. Briefly, Int$^+$ and Int$^-$ strains from *cis* and *trans* systems were grown and induced as previously described, and total RNA was prepared. RNAs were polyA-tailed using *E. coli* Poly(A) Polymerase (New England BioLabs, Ipswich, MA). 5′ and 3′ RACE experiments were then done using Invitrogen 5′ or 3′ RACE kits (Cat# 18374058 or 18373019), or the Takara SMARTer 5′/3′ kit (Cat# 634859), following the kit protocols. Reverse transcriptase was used to synthesize cDNAs, which were then used as template for PCR amplification using GSP (gene specific primers IDT6074 and IDT6070 to *ltrB* 5′ or 3′ exon, respectively) along with a kit universal primer annealing to the synthesized cDNA. 5′ and 3′ RACE PCR products were run out on a 1.2% agarose gel, and the bands were excised, gel purified, and sequenced directly (Eton Bioscience, San Diego, CA) or cloned into pGEM-T vector (Promega), and then sequenced, to confirm their identity.

## Acknowledgements

We thank Alan Lambowitz for providing DNA for cloning the EcI5 group II intron. This work was supported by National Institutes of Health Grants GM39422 and GM44844 (to MB).

## Additional information

### Funding

| Funder | Grant reference number | Author |
| --- | --- | --- |
| National Institutes of Health | GM39422 | Marlene Belfort |
| National Institutes of Health | GM44844 | Marlene Belfort |

The funders had no role in study design, data collection and interpretation, or the decision to submit the work for publication.

### Author contributions

Guosheng Qu, Conceptualization, Data curation, Formal analysis, Supervision, Investigation, Methodology, Writing—original draft, Writing—review and editing; Carol Lyn Piazza, Data curation, Formal analysis, Investigation, Methodology, Writing—original draft, Writing—review and editing; Dorie Smith, Data curation, Formal analysis, Investigation, Methodology, Writing—review and editing; Marlene Belfort, Conceptualization, Formal analysis, Supervision, Funding acquisition, Writing—original draft, Project administration, Writing—review and editing

### Author ORCIDs

Guosheng Qu (ID) http://orcid.org/0000-0002-0062-1929
Marlene Belfort (ID) http://orcid.org/0000-0002-1592-5618

### Decision letter and Author response

Decision letter https://doi.org/10.7554/eLife.34268.023
Author response https://doi.org/10.7554/eLife.34268.024

## Additional files

### Supplementary files

• Supplementary file 1. Supplementary Table 1: Plasmids. Describes plasmids used in this study. Supplementary Table 2: DNA oligonucleotides. Describes DNA oligonucleotides used in this study. Supplementary Table 3: Primers used in qRT-PCR. Describes the primers used for qRT-PCR.
DOI: https://doi.org/10.7554/eLife.34268.020

• Transparent reporting form
DOI: https://doi.org/10.7554/eLife.34268.021

### Data availability

All data generated or analysed during this study are included in the manuscript and supporting files.

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
