## [Decision Letter]

Thank you for submitting your article "Group II intron inhibits conjugative relaxase expression in bacteria by mRNA targeting" for consideration by *eLife*. Your article has been reviewed by three peer reviewers, and the evaluation has been overseen by Timothy Nilsen as Reviewing Editor and Gisela Storz as the Senior Editor. The following individuals involved in review of your submission have agreed to reveal their identity: Alan M Lambowitz (Reviewer #1); Aaron Robart (Reviewer #3).

The reviewers have discussed the reviews with one another and the Reviewing Editor has drafted this decision to help you prepare a revised submission.

There was a consensus that this work reports for the first time a new pathway that contributes to the exclusion of group II introns from essential genes. All the reviewers felt that it was, in principle, suitable for publication in *eLife*. Nevertheless, some reviewers raised concerns, which if dealt with would strengthen the paper. These issues should be addressed as thoroughly as possible. Because the comments of the reviewers cannot be condensed into a short list of required revisions, the entire reviews are attached.

Reviewer #1:

This manuscript addresses the question of why mobile group II introns inserted within protein-coding genes are rare in bacteria, possibly providing clues as to the evolutionary driving force for the evolution of group II introns into spliceosomal introns. The paucity of group II introns in bacterial protein-coding genes had previously been attributed to slow rates of RNA splicing combined with coupled transcription/translation leading to impaired gene expression and accumulation of translational errors, thereby necessitating the evolution of the nuclear envelope to separate transcription from translation (e.g., Martin and Koonin, 2006). Qu et al. identify a very interesting additional factor, the targeting of mRNA sequence by reverse splicing and spliced-exon reopening reactions of excised group II intron RNAs. Although mobile group II introns are well known to reverse splice and cleave DNA target sequences in vivo, physiologically relevant RNA-guided targeting of RNA sequences in vivo had not been described previously. Thus, this is an interesting and important manuscript, which after satisfactory responses and revisions, should be very suitable for publication in *eLife*.

1) Abstract, – the sentence beginning "Here…" and continuing "inhibits relaxase host-gene expression" may be confusing for a general audience and should be rephrased to make clear at the outset that the intron is present within the relaxase gene and that "host gene" here is referring to the gene in which the intron resides and not genes in the host that harbors the mobile element. Similar changes might help the last paragraph of the Introduction.

2) The Introduction should make reference to other factors that may contribute to the paucity of group II introns in bacterial protein-coding genes and to the evolutionary driving force for the evolution of group II into spliceosomal introns in eukaryotes, such those discussed in Martin and Koonin (2006) and Truong et al., 2015), as well as the authors' previous work on the contribution of nonsense-mediated decay in eukaryotes. The reactions found here are quite interesting but unlikely to be the sole factors that account for the distribution of bacterial group II introns or drove their evolution into spliceosomal introns.

3) Figure 1B – the authors should indicate in the legend whether the data shown are from the same blot after stripping and reprobing.

4) Figure 1C – it would have been helpful to have a protein standard in the western blots to control for differences in efficiency of transfer to the membrane.

5) Figure 1D – It is not clear how greater difference in mRNA levels between Int^-^ and Int^+^ cells measured by Northern hybridization or RT-qPCR can be accounted for by difference in reference RNA standard unless one of the reference RNA standards is changing as a result of the intron being present (in which case it should not have been a reference RNA standard). Perhaps the RT-qPCR is simply more quantitative than the Northern blots, e.g., due to uneven transfer of RNA to the membrane.

6) Figure 1—figure supplement 2 - The sequence shown for the pre-mRNA in panel A appears to be the sequence of the cDNA. Given the location of the primer shown in the figure, it is not clear why the cDNA sequence extends beyond a chain terminating dideoxy T in the cDNA sequence just upstream of the primer. A more descriptive figure legend describing what was done in the experiment would be helpful.

7) – Figure 1—figure supplement 4 - it would be helpful to show the UV spectrum of the gradient in the figure supplement so that readers can evaluate the cutoff from polysome enrichment. I'm also curious as to whether one can detect polysomes translating the intron ORF in unspliced precursor RNA. What are the lanes at the left labeled "+" and "-"? Labels for the top and bottom of the gradient might be helpful.

8) Subsection “A group II intron inhibits gene expression at the mRNA level”, second paragraph. The authors seem to be assuming that the relatively small decreases in the rates of transcription and splicing would result in proportional decreases in mRNA levels, so that the observed larger decrease must reflect intron RNA targeting. However, it seems possible that the small decreases in rates of transcription and splicing are cumulative and synergistic and result in disproportionate decreases in mRNA levels.

9) Figure 2B – the splicing efficiency of the EcI5 intron in the GFP construct appears to be substantially lower than that of the Ll.LtrB intron in the relaxase gene construct and could thus make a substantially higher contribution to decreased mRNA levels with this intron.

10) Figure 2 data not shown – the group IIC intron BhI1 is mentioned in the text as being tested for effect of mRNA levels in parallel with EcI5, but data for this intron are not shown in Figure 2, which shows data only for EcI5. The degree of inhibition of gene expression by a group IIC intron inserted within a gene is of particular interest because the EBS/IBS interactions are very short (EBS1/IBS1, 4 bp; EBS3/IBS2, 1 bp) compared to group IIA and IIB introns, yet these introns are a prime example of group II introns that avoid protein coding genes (by inserting downstream of transcription terminators).

11) Figure 5 – the trans expression system data and controls are excellent for establishing the type of RNA-guided mechanism proposed by the authors. It might be relatively easy to extend that case by similar trans experiments for the EcI5 intron in *E. coli*, perhaps with fewer controls needed for this second example. The generality of the mechanism would then be supported by results for two different introns in two different bacteria. Subsection “Intron-mRNA interaction inhibits gene expression”, first paragraph – I suggest changing to "Similar to a previous study in yeast,…"

12) Figure 5C and D – Does the decrease in mRNA levels return when the mutant EBS is paired with the complementary mutant IBS? I may be missing something here, but I'm unclear how the construct with the mutant IBS splices to produce mature mRNA. It's possible that the mutant IBS1 construct has been paired with a complementary EBS1 in cis, but I can't easily find that information.

13) Figure 5—figure – supplement 1B – is it possible to add a panel showing that splicing of the RT and EN mutant construct occurs at the same level as the wild-type protein? A concern is that these mutant proteins are not produced at wild-type levels and this deficiency or partial inactivation by the mutations may decrease splicing, which would in turn lead to decreased mRNA levels, thereby complicating the interpretation of the results.

14) Figure 6 – although the interpretation of the Northern hybridizations is likely correct, they are not sufficient to characterize this complex set of RNA products. This should be done by RNA-seq, which is needed to verify the proposed junctions.

15) Figure 7 – do RNA products I-IV reappear when the mutant IBS is paired with a comp.

Reviewer #2:

In this manuscript, Qu et al. provide the first evidence for group II introns inhibiting gene expression through mRNA targeting. This is an exciting discovery and provides an explanation for the frequency and distribution of group II introns in prokaryotic and eukaryotic genomes. Specifically, Qu et al. found that the EBS-IBS interactions found in group II introns are responsible for binding to target mRNAs through Watson-Crick pairing and then cleaving/reverse splicing these mRNAs to reduce gene expression. This finding also makes biochemical sense given the in vitro propensity of group II introns to efficiently bind to RNA/DNA substrates through EBS-IBS interactions. The authors also tested their hypothesis using group II introns corresponding to the major phylogenetic classes of IIA, IIB, and IIC introns. However, I was curious as to why the data for the IIC intron B.h.I1 was not shown in Figure 2B. This also has major implications for the evolution of group II introns into spliceosomal introns. In order to maintain expression, the EBS-IBS interactions were lost from the actual introns themselves and instead incorporated into the spliceosome. This may provide a biochemical rationale for the evolutionary transition from a cis-splicing system as seen in group II introns into a trans-system in the spliceosome. Overall, the authors provided thorough in vivo experimental evidence that group II introns inhibit gene expression and I highly recommend publication of this manuscript in *eLife*.

Reviewer #3:

It is well established that bacteria group II introns favor insertions into other selfish elements, such as transposons, and avoid essential host genes. This distribution has been rationalized by group II introns seeking out "safe harbors" in bacterial genomes to avoiding deleterious effects on the host, thus ensuring their survival. However, no substantial biochemical or genetic evidence has been published to show that insertion of a group II intron into essential host genes has unfavorable outcomes. Here Qu and colleagues fill this critical knowledge gap using the well-established group IIA L.l.LtrB model system to investigate how group II intron splicing, and reverse splicing, dictates mRNA accumulation levels. Furthermore, this study was performed in the natural *Lactococcus lactis* host environment, strengthening its significance. The major finding of this manuscript is that group II intron reverse splicing, normally targeted to DNA for initiation of mobility reactions, also targets mRNA and significantly reduces total mRNA accumulation levels.

In principle reverse splicing can occur at both the DNA and RNA level. The DNA mobility reaction pathways are well known; however, before this study it was unclear if the competing RNA reverse-splicing reaction occurred at significant levels what the consequences of this reaction would be. In this manuscript, Qu and Belfort present exciting evidence that RNA level group II intron reverse splicing does indeed occur, but causes the phosphodiester bond connecting the two spliced exons to be opened up, leading to subsequent degradation of the message. The authors characterized this mRNA reduction using thoughtful controls to separate splicing from mobility effects, show that mRNA reduction has significant effects on conjugation efficiency (function encoded by the mRNA), and provide both in vitro and in vivo mechanistic insight. The data are clearly presented, the conclusions are supported, and the significance level is appropriate for *eLife*. I strongly recommend publication.

---

## [Author Response]

Reviewer #1:[…] 1) Abstract, the sentence beginning "Here…" and continuing "inhibits relaxase host-gene expression" may be confusing for a general audience and should be rephrased to make clear at the outset that the intron is present within the relaxase gene and that "host gene" here is referring to the gene in which the intron resides and not genes in the host that harbors the mobile element. Similar changes might help the last paragraph of the Introduction.

Changes have been made as suggested in the Abstract, and the last paragraph of the Introduction.

2) The Introduction should make reference to other factors that may contribute to the paucity of group II introns in bacterial protein-coding genes and to the evolutionary driving force for the evolution of group II into spliceosomal introns in eukaryotes, such those discussed in Martin and Koonin (2006) and Truong et al., 2015), as well as the authors' previous work on the contribution of nonsense-mediated decay in eukaryotes. The reactions found here are quite interesting but unlikely to be the sole factors that account for the distribution of bacterial group II introns or drove their evolution into spliceosomal introns.

Changes have been made to the third paragraph of the Introduction as suggested. At the beginning of the paragraph, we added “Distribution of group II introns is different in bacteria and eukaryotes.” At the end of the paragraph, we added “In addition, group II introns are absent in eukaryotic nuclear genomes although they are ancestrally closely related to spliceosomal introns (Lambowitz and Belfort, 2015; Novikova and Belfort, 2017; Zimmerly and Semper, 2015). Factors, including nucleus-cytosol compartmentalization, intracellular magnesium concentrations, and interactions between the intron and spliced mRNA, have been conjectured to be the evolutionary drivers for the evolution of group II into spliceosomal introns in eukaryotes (Martin and Koonin, 2006; Qu et al., 2014; Truong et al., 2015). “

3) Figure 1B – the authors should indicate in the legend whether the data shown are from the same blot after stripping and reprobing.

In the legend of Figure 1B, we added “from the same blot after stripping and reprobing”.

4) Figure 1C – it would have been helpful to have a protein standard in the western blots to control for differences in efficiency of transfer to the membrane.

Portions of the coomassie stained gels before and after transfer now are shown in Figure 1C, indicating equivalent efficiencies of transfer.

5) Figure 1D – It is not clear how greater difference in mRNA levels between Int^-^ and Int^+^ cells measured by Northern hybridization or RT-qPCR can be accounted for by difference in reference RNA standard unless one of the reference RNA standards is changing as a result of the intron being present (in which case it should not have been a reference RNA standard). Perhaps the RT-qPCR is simply more quantitative than the Northern blots, e.g., due to uneven transfer of RNA to the membrane.

At the end of the first paragraph in Results, we added “or by the inherent difference between the two techniques in determining RNA levels.”

6) Figure 1—figure supplement 2 - The sequence shown for the pre-mRNA in panel A appears to be the sequence of the cDNA. Given the location of the primer shown in the figure, it is not clear why the cDNA sequence extends beyond a chain terminating dideoxy T in the cDNA sequence just upstream of the primer. A more descriptive figure legend describing what was done in the experiment would be helpful.

The sequences shown in Figure 1—figure supplement 2 have been replaced with RNA sequences. Description of reverse transcription terminations in the figure legend have been modified as suggested.

7) – Figure 1—figure supplement 4 - it would be helpful to show the UV spectrum of the gradient in the figure supplement so that readers can evaluate the cutoff from polysome enrichment. I'm also curious as to whether one can detect polysomes translating the intron ORF in unspliced precursor RNA. What are the lanes at the left labeled "+" and "-"? Labels for the top and bottom of the gradient might be helpful.

A UV spectrum of the gradient now has been incorporated into the figure. Labels for "+" and "-" have been defined in the legend and labels for the top and bottom of the gradient have been added to the figure. Regarding the translation of the intron ORF in unspliced intron precursor in which the reviewer is interested, our Northern hybridization (using the same blot after stripping of mRNA signals) shows that unspliced intron is present in all the polysomal fractions.

8) Subsection “A group II intron inhibits gene expression at the mRNA level”, second paragraph. The authors seem to be assuming that the relatively small decreases in the rates of transcription and splicing would result in proportional decreases in mRNA levels, so that the observed larger decrease must reflect intron RNA targeting. However, it seems possible that the small decreases in rates of transcription and splicing are cumulative and synergistic and result in disproportionate decreases in mRNA levels.

We agree that concurrent small decreases in rates of transcription and splicing could account for some decreases in mRNA levels. To make our statements less focused on the effect of a single factor on the mRNA loss, we have modified the following sentences in the second paragraph of Results.

Second paragraph: “By performing both reverse transcription and qRT-PCR analyses we determined that the decrease in mRNA was not due to group II intron-promoted reduction in transcription rate or splicing (Figure 1—figure supplement 1, 2).” changed to “By performing both reverse transcription and qRT-PCR analyses we determined that the decrease in mRNA was not simply due to group II intron-promoted reduction in transcription rate (Figure 1—figure supplement 1).”

Third paragraph 3: “Thus, it does not appear that a splicing deficiency resulted in the mRNA reduction.” changed to “Thus, it does not appear that a splicing deficiency *alone*, resulted in the mRNA reduction.”

9) Figure 2B – the splicing efficiency of the EcI5 intron in the GFP construct appears to be substantially lower than that of the Ll.LtrB intron in the relaxase gene construct and could thus make a substantially higher contribution to decreased mRNA levels with this intron.

We accept this point and have indicated splicing efficiencies in the legend to Figure 2. We noted that splicing of either intron from these reporter constructs was at relatively low efficiency (EcI5: 17%; BhI1: 25%), which could be attributed to an inherent property of these introns in *E. coli,* or to improper IEP protein folding accompanying overexpression. Sharply decreased mRNA levels in the presence of the intron are nevertheless readily apparent.

10) Figure 2 data not shown – the group IIC intron BhI1 is mentioned in the text as being tested for effect of mRNA levels in parallel with EcI5, but data for this intron are not shown in Figure 2, which shows data only for EcI5. The degree of inhibition of gene expression by a group IIC intron inserted within a gene is of particular interest because the EBS/IBS interactions are very short (EBS1/IBS1, 4 bp; EBS3/IBS2, 1 bp) compared to group IIA and IIB introns, yet these introns are a prime example of group II introns that avoid protein coding genes (by inserting downstream of transcription terminators).

We have included the data for the BhI1 intron into Figure 2C, and made changes accordingly in the figure legend and main text.

11) Figure 5 – the trans expression system data and controls are excellent for establishing the type of RNA-guided mechanism proposed by the authors. It might be relatively easy to extend that case by similar trans experiments for the EcI5 intron in E. coli, perhaps with fewer controls needed for this second example. The generality of the mechanism would then be supported by results for two different introns in two different bacteria. Subsection “Intron-mRNA interaction inhibits gene expression”, first paragraph – I suggest changing to "Similar to a previous study in yeast,…"

We appreciate the reviewer’s idea but such a parallel study is out of the scope of this work. Similar result in yeast is referred to in the Results: “Using the *trans*-expression system, we validated that the intron-mRNA interaction requires EBS-IBS base-pairing, as shown previously in yeast (Qu et al., 2016)”

12) Figure 5C and D – Does the decrease in mRNA levels return when the mutant EBS is paired with the complementary mutant IBS? I may be missing something here, but I'm unclear how the construct with the mutant IBS splices to produce mature mRNA. It's possible that the mutant IBS1 construct has been paired with a complementary EBS1 in cis, but I can't easily find that information.

We have redesigned the IBS and EBS mutants (m*), such that they are compensatory, where the mutated IBS within the relaxase mRNA and the mutated EBS within the intron complement each other. While basically reproducing the result for IBS and EBS mutants that was shown in the previous submission, we observed the decrease in mRNA levels returns in the compensatory mutant. These results now have been described in Figure 5—figure supplement 2 and Figure 7, and in the accompanying text.

13) Figure 5—figure supplement 1B – is it possible to add a panel showing that splicing of the RT and EN mutant construct occurs at the same level as the wild-type protein? A concern is that these mutant proteins are not produced at wild-type levels and this deficiency or partial inactivation by the mutations may decrease splicing, which would in turn lead to decreased mRNA levels, thereby complicating the interpretation of the results.

Splicing of the intron in the *in trans* system is indicated by production of the smRNA. smRNA occurs for the RT and EN mutant constructs at almost the same level as the wild-type protein (Figure 5—figure supplement 1B). We have added a sentence to the legend of Figure 5—figure supplement 1B: “Intron splicing, which was indicated by the production of the smRNA, was not substantially affected in these mutants although mRNA levels were reduced, indicating no major splicing defects in these mutants”.

14) Figure 6 – although the interpretation of the Northern hybridizations is likely correct, they are not sufficient to characterize this complex set of RNA products. This should be done by RNA-seq, which is needed to verify the proposed junctions.

By designing 5’- and 3’-RACE experiments followed by DNA sequencing, we have validated the presence of RNAs 1-4 (Figure 6—figure supplement 2). The results are all perfectly consistent with those from Northern blotting.

15) Figure 7 – do RNA products I-IV reappear when the mutant IBS is paired with a comp.

Using the redesigned IBS and EBS mutants (m*) (see response to comment 12), we redid Northern blotting analysis and it reproduces the result for the original IBS and EBS mutants and shows the reappearance of products 1-4 in the compensatory mutant. These results have been included in Figure 5—figure supplement 2 and Figure 7 and changes have been made to the figure legend and relevant paragraph in Results.

Reviewer #2:In this manuscript, Qu et al. provide the first evidence for group II introns inhibiting gene expression through mRNA targeting. This is an exciting discovery and provides an explanation for the frequency and distribution of group II introns in prokaryotic and eukaryotic genomes. Specifically, Qu et al. found that the EBS-IBS interactions found in group II introns are responsible for binding to target mRNAs through Watson-Crick pairing and then cleaving/reverse splicing these mRNAs to reduce gene expression. This finding also makes biochemical sense given the in vitro propensity of group II introns to efficiently bind to RNA/DNA substrates through EBS-IBS interactions. The authors also tested their hypothesis using group II introns corresponding to the major phylogenetic classes of IIA, IIB, and IIC introns. However, I was curious as to why the data for the IIC intron B.h.I1 was not shown in Figure 2B. This also has major implications for the evolution of group II introns into spliceosomal introns. In order to maintain expression, the EBS-IBS interactions were lost from the actual introns themselves and instead incorporated into the spliceosome. This may provide a biochemical rationale for the evolutionary transition from a cis-splicing system as seen in group II introns into a trans-system in the spliceosome. Overall, the authors provided thorough in vivo experimental evidence that group II introns inhibit gene expression and I highly recommend publication of this manuscript in eLife.

The data of BhI1 intron now have been included (Figure 2C).